# Solving Blind Non-linear Forward and Inverse Problem for Audio Applications

## Abstract

We propose a unified framework to address the *blind forward and inverse problems* in audio domain, where the objective is to estimate either the function or the input signal solely from the observed output, without access to the other. We formally define *forward operators* — mapping input to output signals — and formulate both problems within a probabilistic framework. For the blind forward problem, we design an architecture that utilizes a reference encoder to extract features from the reference signal, enabling the main operator to approximate arbitrary forward operators systematically composed via algebraic representations. For the blind inverse problem, we employ a conditional diffusion model conditioned on features from the pretrained reference encoder and augment the generation process using *twisted particle filtering* technique leveraging the approximated operator in the forward problem. We validate our framework on *zero-shot audio effect modeling* and *speech enhancement*. The experiments show that our approach replicates both simple and complex audio effects, generalizes under distribution mismatches, and effectively enhances noisy full-band audio across diverse effects and real-world scenarios. Codes are available at `https://t.ly/n1luk`, with audio samples at `https://t.ly/dBUhF`

## 1 Introduction

In both classical digital signal processing (DSP) and modern deep learning-based approaches, a central assumption is the existence of a 'function' that maps input signals to their corresponding outputs. While the classical DSP decomposes the speech into filters and sound source according to the source-filter theory (McAulay & Quatieri, 1986; Oppenheim, 1999), deep learning-based method leverages neural networks to approximate more complex mappings between *input-output pairs* with a supervised manner even without knowledge on the characteristics of the mappings.

Still, recovering either the function or its input *solely from the output signal* remains challenging. Common approaches to estimate the functional form is to simplify or precondition the functional form or restrict it to specific types, collapsing the problem in parameter estimation or classifications (Engel et al., 2020; Colonel & Reiss, 2021; Lee et al., 2022b; Colonel et al., 2022; Peladeau & Peeters, 2024; Guo & McFee, 2023; Lee et al., 2023b; Rice et al., 2023; Take et al., 2024). However, these methods are only able to handle limited and simplified types, often difficult to be generalized to the wild setting. To reconstruct the input signals from the output signals, discriminative models learn to either directly predict the input signal from the given output signal or find a mask that results in a cleaned version of the output; thereby solving speech enhancement or signal restoration task. Recently, generative models are also one of the choices to recover the input signal by using the output signal as an auxiliary information in the generation process (Abdulatif et al., 2024; Serrà et al., 2022; Richter et al., 2023; Lemercier et al., 2023). These models are particularly effective in handling diverse types of degradation without relying on specific noise models. However, they ignore the connection between the clean and noisy signals, hence require large clean and noisy signal pairs to handle diverse degradation types.

In this work, we introduce a unified framework to solve the *blind forward and inverse problems in the audio domain*, focusing on estimating either the function or the input signal from the observed output. After the definition of forward operators and each problems are rigorously formalized, we develop a framework to approximate arbitrary forward operators constructed by the algebraic

approach. For the blind inverse problem, we leverage a conditional diffusion model conditioned by the pre-trained reference encoder, and apply a *twisted particle filtering* method using the pre-trained estimated forward operator. The effectiveness of the proposed method is empirically demonstrated via zero-shot audio effect modeling and speech enhancement.

## 2 RELATED WORKS

**Zero-shot Audio Effect Modeling**   We address zero-shot audio effect modeling as a representative example of the blind forward problem. A common approach to this task involves simplifying and preconditioning the functional form using DSP-based domain knowledge and strong inductive biases. For instance, the problem is often reduced to estimating intermediate features such as parameters (Engel et al., 2020; Colonel & Reiss, 2021; Lee et al., 2022b; Colonel et al., 2022; Peladeau & Peeters, 2024), impulse responses (Steinmetz et al., 2021; Lee et al., 2023a), or combinations of audio effects (Guo & McFee, 2023; Lee et al., 2023b; Rice et al., 2023; Take et al., 2024). Other approaches focus on specific tasks, such as acoustic scene transfer (Im & Nam, 2024), impulse response learning (Steinmetz et al., 2021), or modeling single-type effects (Chen et al., 2024).

**Speech Enhancement**   Speech enhancement is a quintessential example of the blind inverse problem in the audio domain. In deep learning-based approaches, the primary method involves using discriminative or GAN-based models to estimate a mask in either the time or time-frequency domain (Luo & Mesgarani, 2019; Lu et al., 2022b; Abdulatif et al., 2024; Choi et al., 2019). Recently, diffusion models have emerged as a promising alternative for speech enhancement. For example, Serrà et al. (2022) conditioned score-based models on noisy signals, while Welker et al. (2022); Richter et al. (2023); Lemercier et al. (2023) designed a forward and reverse process coherent to the enhancement process. However, these prior methods primarily treated noisy signals as auxiliary information, failing to fully exploit the functional relationship between clean and noisy signals.

**Diffusion-based Inverse Problem**   Recently, diffusion-based method gained prominence in solving inverse problems. Since the score function of the posterior distributions appears as a guidance term during the diffusion steps, several works approximate the terms using known measurement operators (Song et al., 2021; Chung et al., 2023b; Ho et al., 2022). Some lines of works factorize the linear operator using SVD (Kawar et al., 2022) or use pseudo inverse to approximate the operator (Song et al., 2023). In cases where the operator is unknown, referred to as a *blind* setting, prior works estimate the parametrized operators during diffusion process such as (Chung et al., 2023a; Murata et al., 2023). In audio domain, the inverse problem is solved to remove audio effects such as Moliner et al. (2024); Lemercier et al. (2024). Recently, several works integrate Sequential Monte Carlo (SMC) methods with diffusion models to enhance the conditional generation.Cardoso et al. (2023); Wu et al. (2023); Dou & Song (2024); Nazemi et al. (2024)

## 3 DEFINITIONS, NOTATIONS, AND THE PROBLEM FORMULATION

Let $K \subseteq \mathbb{R}^T$ be a signal space, where $T$ denotes the signal length. Consider a mapping $\mathcal{A} : K \to K$ that defines the system $y = \mathcal{A}(x)$ for $x \in K$. We refer to the input signal $x$ as the *dry signal* and the output signal $y$ as the *wet signal*. To analyze such system, we introduce the following function class of our interest.

**Definition 1** (Forward Operator)**.** For a signal space $K \subseteq \mathbb{R}^T$, A *forward operator* $\mathcal{A}$ is a continuous bounded function $\mathcal{A} : K \to K$. The set of all forward operators is denoted by

$$C_b(K) = \{\mathcal{A} : K \to K, \mathcal{A} \text{ is continuous}, \|\mathcal{A}(x)\| \le M\|x\|, \forall x \in K \text{ for some } M > 0 \in \mathbb{R}\} \quad (1)$$

The advantage of this definition is the composition of the two forward operators, namely $\mathcal{A} \circ \mathcal{A}'$ for any $\mathcal{A}, \mathcal{A}' \in C_b(K)$, is well-defined and also lies in $C_b(K)$. With the continuity assumption, the wet signal is assumed not to be abruptly changed by the small perturbation in the dry signal. Additionally, the boundedness condition ensures that the output signal does not become unbounded when the input is properly normalized.

**Examples of the Forward Operator**   An important class of forward operators is *audio effects*, which process dry speech or music to produce wet signals with modifications such as equalization, reverberation, or filtering. Another key class is *degradations*, although not mutually exclusive with

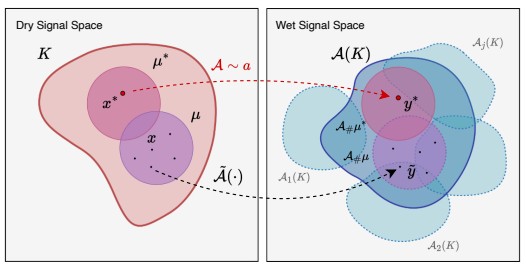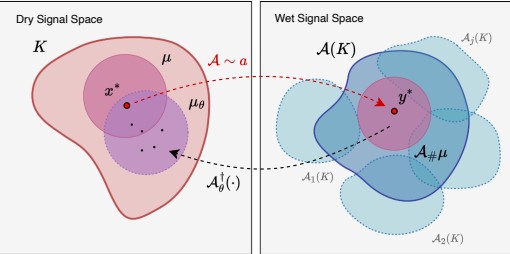

Figure 1: A reference signal pair $(x^*, y^*)$ is generated by sampling $x^* \sim \mu^*$, $\mathcal{A} \sim a$, and $y^* = \mathcal{A}(x^*)$. (Left) Blind forward problem: $\tilde{\mathcal{A}}_\theta[y^*] \approx \mathcal{A}$ is approximated so that the push-forward measure of $\mu$ induced by $\mathcal{A}$ and $\tilde{\mathcal{A}}$ coincides, and (Right) Blind inverse problem : the inverse mapping $\tilde{\mathcal{A}}_\phi[y^*]$ is approximated so that its push-forward measure is matched to $\mu$

audio effects, which commonly encountered in speech enhancement tasks, where a clean speech is corrupted by noise, distortion, audio codecs, or reverberation. In this case, the noisy speech $y$ is a wet signal produced by an unknown degradation operator $\mathcal{A}$, where $y = \mathcal{A}(x)$ and $x$ is the clean signal. In particular, many forward operators in the audio domain are often highly *nonlinear and even non-differentiable* (e.g. audio codec). In this work, we equally treat the audio effect and degradation as forward operators without any distinction.

### 3.1 PROBABILISTIC FORMULATION

Although the forward operators are fully deterministic, we adopt a probabilistic formulation for the following advantages: 1) it reflects the sampling nature of the signal dataset and the randomly constructed operator, 2) it parallelizes the forward and inverse problem in a generative framework, and 3) it addresses ill-posed problems, such as irreversible operations (e.g., lowpass filtering) that lose information in reconstruction and lead to one-to-many mappings. In such cases, a deterministic inverse may not exist; however, a probabilistic approach allows for finding the most probable solution. This parallels the Kantorovich formulation, which overcomes the limitations of Monge approach in transportation theory (Villani, 2008).

Let $\mu, a$ be probabilistic measures of the dry signals and forward operators on $K$ and $C_b(K)$, respectively. If we fix any forward operator $\mathcal{A}$, this induces a probability measure on the wet signals via the *push-forward measure* by $y \sim \mathcal{A}_\# \mu$. Accordingly, we propose our objective as a 'distribution matching sense' between this push-forward measure and the target distribution for each problem.

**Definition 2** (Forward / Inverse Problem). Let $\mu^*$ be a probability measure of the reference dry signals. For a reference pair $(x^*, y^*)$ generated by $y^* = \mathcal{A}(x^*)$, The *blind forward problem* aims to approximate a neural network $\tilde{\mathcal{A}}_\theta[y^*]$ conditioned only on $y^*$ to $\mathcal{A}$ so that $\|\tilde{\mathcal{A}}_\theta[y^*] - \mathcal{A}\|_{L_2(\mu)} \to 0$. Therefore, the objective is given as the distance between the push-forward measures of dry signals induced by $\mathcal{A}$ and $\tilde{\mathcal{A}}_\theta$:

$$\text{(BFP)} \quad \min_\theta \mathbb{E}_{\mathcal{A} \sim a, x^* \sim \mu^*} \left[ \mathcal{W}_2(\tilde{\mathcal{A}}_\theta[y^*]_\# \mu, \mathcal{A}_\# \mu) \right], \quad y^* = \mathcal{A}(x^*) \tag{2}$$

where $\mathcal{W}_2$ denotes the 2-Wasserstein distance.

In contrast, the *blind inverse problem* aims to approximate a neural network $\tilde{\mathcal{A}}_\phi[y^*]$ conditioned only on $y^*$ so that $\|\tilde{\mathcal{A}}_\phi[y^*] \circ \mathcal{A} - id\|_{L_2(\mu)} \to 0$, where $id$ is the identity map. Therefore, we minimize the following loss:

$$\text{(BIP)} \quad \min_\phi \mathbb{E}_{\mathcal{A} \sim a, x^* \sim \mu^*} \left[ \mathcal{W}_2((\tilde{\mathcal{A}}_\phi[y^*] \circ \mathcal{A})_\# \mu, \mu) \right], \quad y^* = \mathcal{A}(x^*) \tag{3}$$

It is noteworthy that the approximation of the operator $\tilde{\mathcal{A}}_\theta$ has a dependence on the dry signal space $\mu$. In other words, the approximation is only guaranteed on the dry signals drawn from $\mu$. We examine its generalization performance in the experiment.

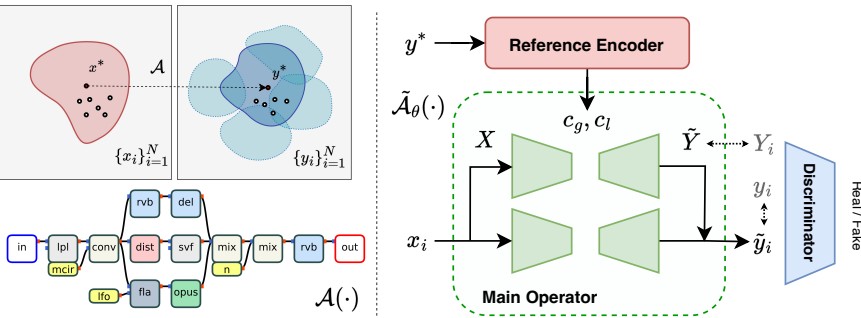

Figure 2: (Left) The reference signal pair $(x^*, y^*)$ and the target signal pairs $\{(x_i, y_i)\}_{i=1}^N$ are generated by the sampled forward operator $\mathcal{A}$ which is represented by a DAG whose structures and parameters are randomized. The cryptic acronyms (e.g., lpl) represent a type of audio effect and are listed in the Appendix C.2. (Right) The reference encoder returns $c_g, c_l$ from the reference signal $y^*$. Then the dry signal $x_i$ and its STFT signal $X$ are processed to return $\tilde{y}_i$.

## 4  SOLVING BLIND FORWARD PROBLEM

Note that we have access only to $y^*$, not to the reference pair $(x^*, y^*)$, for approximating the ground-truth operator $\mathcal{A}$, where $y^* = \mathcal{A}(x^*)$. To address this, the reference encoder extracts informative features from $y^*$, producing a global condition vector $c_g$ and a sequence of local condition vectors $c_l$. Then the main operator network transforms $K$ i.i.d. dry signals $\{x_i\}_{i=1}^K$ drawn from the distribution $\mu$ into wet signals $\{y_i\}_{i=1}^K$ by conditioning on $c_g, c_l$. Finally, a discriminator is applied to eliminate artifacts and improve the quality of the generated signal.

### 4.1  ARCHITECTURE CHOICES

**Reference Encoder**  We adopt the encoder part of MTFAA-Net from Zhang et al. (2022), which has demonstrated strong performance in speech enhancement. In speech enhancement architectures, the encoder typically extracts features related to non-speech components, allowing the decoder to separate them from speech. In our framework, we leverage the reference encoder to extract informative features from $y^*$ that help for the main operator to approximate arbitrary forward operators. The extracted feature $z$ is then transformed into a global condition $c_g \in \mathbb{R}^{d_g}$ and a local condition $c_l \in \mathbb{R}^{d_l \times N}$ by respective conditioning module, where $d_g$ and $d_l$ are embedding dimensions of each, and $N$ is the number of tokens of $c_l$.

**Main Operator Network**  We employ the U-Net architecture from the Imagen text-to-image model (Saharia et al., 2022) as our main operator network. In text-to-image diffusion models, the network is conditioned on $t$, representing the noise variance, and $c$, the text embedding from a pre-trained encoder. We hypothesize that $t$ provides *global* information on noise scale, while $c$ adjusts *local image details* via cross attention (Hertz et al., 2022; Ramesh et al., 2022). Based on this, we replace time conditioning $t$ with a global condition $c_g$ and text embedding $c$ with a local condition $c_l$ from the reference encoder.

Additionally, we process the dry signal in both waveform and spectrogram domains, following (Rouard et al., 2023; Défossez, 2021). This dual-domain approach significantly enhances model performance, as certain effects are more easily detected in one domain than the other. Finally, we concatenate the wet reference signal $y^*$ with the dry target signal $x$ across the channel axis. This is particularly useful for replicating additive noise components, as they align perfectly in the temporal or time-frequency domain.

**Discriminator**  To remove possible artifacts from the predicted signal, we apply a multi-resolution discriminator (MRD) Lee et al. (2022a). The MRD consists of multiple sub-discriminators, each processing magnitude spectrograms at different resolutions. As we found that incorporating a multi-period discriminator (MPD) deteriorates the overall performance, MRD is only employed.

## 4.2 TRAINING OBJECTIVE

We propose the following training objectives to achieve the minimization of the Equation 2:

$$L_G(G; D) = \sum_{i,j} \left[ \|\tilde{\mathcal{A}}_\theta[y_j^*](x_i) - \mathcal{A}_j(x_i)\|^2 + \lambda_m \mathcal{L}_m(\tilde{\mathcal{A}}_\theta[y_j^*](x_i), \mathcal{A}_j(x_i)) + \lambda_{adv} \mathcal{L}_{adv}(G; D) \right]$$

$$L_D(D; G) = \mathcal{L}_{adv}(D; G)$$

where $\mathcal{L}_m$ is the loss function in the magnitude spectrogram domain, $\lambda_m, \lambda_{adv}$ are some constants, and $\mathcal{L}_{adv}(D; G)$ and $\mathcal{L}_{adv}(G; D)$ are the adversarial loss with the freezed parameters of the model and the discriminator, respectively. Note that we approximate the 2-Wasserstein distance in Equation 2 by that of empirical distributions. In addition, we also add the loss function measured in magnitude spectrogram and adversarial loss to enhance the perceptual quality of the predicted signals The details of each loss function are in the Appendix C.3.

# 5 CONSTRUCTION OF THE FORWARD OPERATOR

Since our model is desired to approximate any general forward operator $\mathcal{A}$, it must be constructed and be able to render the wet signal *on-the-fly* from the given dry signal while training. However, sampling from the function space we defined in Equation 1 is not only too large but also redundant as we do not want pathological operators that may completely obscure the contents information of the dry signal. Therefore, we simulate a forward operator by *composing known and closed-form operators* in the following combinatorial way.

## 5.1 ALGEBRAIC COMPOSITION OF THE FORWARD OPERATOR

From our definition in Equation 1, observe that the addition and multiplication between two operators $\mathcal{A}, \mathcal{A}' \in C_b(K)$ for a signal $x \in K$ are well-defined as :

$$\begin{aligned}
\text{(Addition)} &: (\mathcal{A} + \mathcal{A}')(x) := \mathcal{A}(x) + \mathcal{A}'(x) \\
\text{(Multiplication)} &: (\mathcal{A} \cdot \mathcal{A}')(x) := (\mathcal{A} \circ \mathcal{A}')(x) = \mathcal{A}(\mathcal{A}'(x))
\end{aligned} \tag{4}$$

Hence we can impose a *semiring* structure on the function space $C_b(K)$.

**Definition 3.** A semiring $(R, +)$ is a set equipped with two binary operations $(+, \cdot)$ which satisfy the following: 1) $R$ is associative and commutative under addition. 2) $R$ is associative under multiplication. 3) The multiplication is distributive with respect to the addition.

We define $R = (C_b(K), +, \cdot, \bar{0}, \bar{1})$ as a *composition semiring* equipped with the operation $(+, \cdot)$ with the additive identity $\bar{0} : x \mapsto 0$ and multiplicative identity $\bar{1} : x \mapsto x$. Then we refer to the *rendering* the forward operator $\mathcal{A}$ to the dry signal $x$ simply as a semiring action on the signal space $K$ by evaluation $R \times K \to K : (\mathcal{A}, x) \mapsto \mathcal{A}(x)$, which is approximated by $\tilde{\mathcal{A}}_\theta$.

**Advantages of the Semiring Construction** The introduction of a composition semiring provides several theoretical and practical benefits. First, since each element in the composition semiring uniquely characterizes the combination of basic operators, the sampling of a representation from $R$ is equivalent to the sampling of a complex operator. Random parameters are then assigned to each basic operator to complete the construction of the forward operator. Second, by factorizing terms in $\mathcal{A} \in R$ using distributivity, each basic operator in $\mathcal{A}$ is processed only once to render any dry signal $x$, enabling a sequential rendering from the right-most element of $\mathcal{A}$. This results in a total rendering time of $O(N)$ where $N$ is the number of elements.

**Degeneracy Problem** Prior approaches such as (Rice et al., 2023; Lee et al., 2023b; 2024) are equivalent to find the representation of the forward operator by classifications. However, these methods face a *degeneracy problem*, where different representations yield the same action on signals. For instance, two linear time-invariant (LTI) operators $L, L' \in R$ are commutative under multiplication, resulting in identical actions. Similarly, complementary effects, such as two sequential low-shelf filters with opposite gains, cancel each other out. In contrast, our framework directly approximates the action of $\mathcal{A}$ on $x$ rather than its explicit representation, naturally avoiding degeneracy problem and diverging from prior approaches.

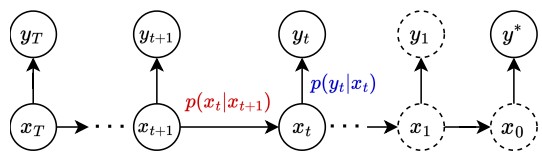

Figure 3: Directed Acyclic Graph (DAG) representation of the Forward Operator. (Left) Single type: this form represents a single element in the set of basic operators $\mathcal{S}$. (Middle) Monolithic type: monomials in $R$. i.e. $\prod \mathcal{A}_i$ where $\mathcal{A}_i \in \mathcal{S}$. (Right) General Type: a general element in $R$.

## 5.2 Properties of the Constructed Forward Operators

In practice, we construct an arbitrary forward operator from the known and closed-form operators, called *basic operators*, detailed in Appendix C.2 for the comprehensive list and technical details. Let $\mathcal{S}$ be a set of the basic operators in $C_b(K)$. A forward operator is formed through summations and multiplications of elements in $\mathcal{S}$, for example, $\mathcal{A} = \sum_j (\prod_i S_i)_j$ for $S_i \in \mathcal{S}$.

**DAG Representation** This construction can be effectively visualized as a *directed acyclic graph (DAG)*, where nodes represent basic operators, as shown in Figure 3 (Lee et al., 2023b; 2024). Summations are represented as parallel compositions, and multiplications as serial compositions. Starting from the input node [in], the DAG is constructed iteratively from the right-most element of $\mathcal{A}$ using serial and parallel compositions, ending at the output node [out]. See Appendix A.1 for details. This DAG representation emphasizes the modularity and structure of the composition semiring, providing an intuitive framework for analysis and implementation.

**Universal Approximation** A natural question arises: "*Can any forward operator in $C_b(K)$ be simulated by our construction?* If so, what necessary conditions on the collection of basic operators $\mathcal{S}$ must be met to achieve such an approximation?" To prove the *universal approximation property*, we imitate the proof of the universal approximation theorem of the single-layer MLPs (Pinkus, 1999). The main idea is to construct $\mathcal{S}$ by linear operators and point-wise nonlinear operators, analogous to affine transformations and activation functions in MLPs (Lin & Pinkus, 1993). The proof is presented in Appendix $A.2$.

## 6 Solving Blind Inverse Problem

### 6.1 State Space Model (SSM) Perspective of the Diffusion Model

**Diffusion Model** A diffusion model (Song et al., 2021; Ho et al., 2020; Karras et al., 2022; Song et al., 2022) perturbs a data distribution by adding Gaussian noise with a scheduled variance $\sigma_t$, forming a stochastic process $x_t = x_0 + \sigma_t z$ where $x_0 \sim \mu, z \sim \mathcal{N}(0, I)$, governed by a forward SDE. The model learns a *denoiser function* $D_\theta(x_t, t)$ to recover $x_0$ from $x_t$, which also estimates the score function of $x_t$ via $\nabla_{x_t} \log p(x_t) = (D_\theta(x_t, t) - x_t)/\sigma_t^2$. The time-reversed SDE, which matches the marginal distribution of the forward SDE at each time step, is solved for data generation.

Figure 4: State Space model. The sequence $x_{T:t}$ and $y_{T:t}$ with the future observation $y_0 = y^*$ are observed at time $t$, while the dashed circles are not observed. $p(x_t|x_{t+1})$ and $p(y_t|x_t)$ are the transition density and observation density of the SSM, respectively.

This process requires the score function $\nabla_{x_t} \log p(x_t)$, approximated using $D_\theta$. Starting from Gaussian noise $x_T$, the reversed SDE iteratively transforms $x_T$ into a clean signal $x_0$.

**Numerical SDE Solvers as SSM** As analytic solutions to the time-reversed SDE are generally intractable, numerical solvers such as Euler-Maruyama (Song et al., 2021), Heun's method (Karras et al., 2022), DPM (Lu et al., 2022a), EI solvers (Zhang & Chen, 2023) are used for generation. These solvers discretize time into finite steps $\{t_i\}_{i=T}^0$ and recursively update $x_t$ as $x_{t-1} = f(x_t, t) + g(t)z_t$ where $z_t \sim \mathcal{N}(0, I)$. Here $f$ and $g$ are one-step updates derived from the forward SDE and depend on the chosen approximation method. This discretization forms a Markov chain over $\{x_i\}_{i=T}^0$.

Let $y^*$ be the given wet signal generated by $y^* = \mathcal{A}(x^*)$. Let $y_t \sim N(y^*, \bar{\sigma}_t I)$ with $y_0 = y^*$ be a observation sequence. Then we form a *state space model (SSM)* by assigning $y_t$ to each $x_t$ of the Markov chain, as shown in Figure 4. Finally, our objective is to sample $x_0 \sim p(x_0|y^*)$ to generate an enhanced speech signal for the given noisy observation $y^*$. To achieve this, we estimate $p(x_0|y_{T:0})$, called *the marginal filtering distribution*, and marginalize over $y_{T:1}$ to compute $p(x_0|y^*)$.

## 6.2 Particle Filtering for the Blind Inverse Problem

**Particle Filtering (PF)** Particle filtering, a variant of the Sequential Monte Carlo (SMC), sequentially updates the distribution $p(x_t|y_{T:t})$, ultimately reaching $p(x_0|y_{T:0})$ at the terminal time (Naesseth et al., 2022; Olsen, 2022; Guarniero et al., 2016; Heng et al., 2020). We apply it to solve the blind inverse problem by computing $p(x_0|y^*)$ based on the previously constructed SSM. The marginal filtering distribution $p(x_t|y_{T:t})$ is factorized into the following recursive alternating steps :

$$\text{(Prediction Step)} : p(x_t|y_{T:t+1}) = \int p(x_t|x_{t+1})p(x_{t+1}|y_{T:t+1})dx_{t+1}$$
$$\text{(Update Step)} : p(x_t|y_{T:t}) = \int p(y_t|x_t)p(x_t|y_{T:t+1})/Z_t \tag{5}$$

where $Z_t$ denotes the normalizing constant. While these integrations have closed-form solutions for specific cases, such as discrete-state or linear Gaussian SSM (e.g., Kalman filter), they are generally intractable for our SSM. Consequently, PF approximates the integration using Monte Carlo sampling with $N$ number of particles $\{x_t^i\}_{i=1}^N$ and corresponding weights $\{w_t^i\}_{i=1}^N$ by :

$$\hat{p}(x_t|y_{T:t+1}) = \sum_{i=1}^N w_{t+1}^i p(x_t^i|x_{t-1}^i),$$
$$\hat{p}(x_t|y_{T:t}) = p(y_t|x_t) \sum_{i=1}^N w_{t+1}^i p(x_t^i|x_{t+1}^i)$$
$$w_t^i \propto w_{t+1}^i \frac{p(y_t|x_t^i)p(x_t^i|x_{t+1}^i)}{g(x_t^i|x_{T:t+1}^i, y_{T:t})}$$

where $g$ is the proposal distribution. The choice of $g = p(x_t|x_{t+1})$ simplifies the weight update to $w_t^i = w_{t+1}^i p(y_t|x_t^i)$.

---

**Algorithm 1:** Twisted Particle Filter

**Input:** $(p_T(x_T), g, p_\theta, y^*, T, N)$
**for** $t \in \{T, ..., 0\}$ **do**
    **for** $i \in \{1, \ldots, N\}$ **do**
        /* Proposal */
        **if** $t = T$ **then**
            Sample
            $x_T^i \sim p_T(x_T)p_\theta(y^*|x_T)$;
            $\tilde{w}_T^i = 1/N$
        **else**
            Sample $x_t^i \sim g(x_t|x_{t+1}^i)$;
            $\tilde{w}_t^i \sim \frac{p(x_t|x_{t+1}^i)}{g(x_t|x_{t+1}^i)} \frac{p_\theta(y^*|x_t^i)}{p_\theta(y^*|x_{t+1}^i)}$
    Normalize weights
    $w_t^i = \tilde{w}_t^i / \sum_{l=1}^n \tilde{w}_t^l$;
    /* Resampling */
    **for** $i \in \{i, \ldots, N\}$ **do**
        $k \sim categorical(\{w_t^i\}_{i=1}^N)$;
        $x_t^i \leftarrow x_t^k$

---

**Twisting Particle Filter** The estimation of $p(x_t|y_{T:t})$ relies on the prediction and update steps *up to time* $t$. However, we incorporate the future observation the future observation $y_0 = y^*$ at time $t$, using *twisted particle filtering*, where prediction and update steps are conjugated by the twisting function $\psi_t(x_t)$, (Olsen, 2022; Zhao et al., 2024). Following Wu et al. (2023); Zhao et al. (2024), we choose the optimal twisting function $\psi(x_t) = p_\theta(y^*|x_t) \approx \mathcal{A}(\hat{x}_0)$, where $\hat{x}_0 \approx \mathbb{E}_{0|t}[x_0|x_t]$, yielded by the Tweedie's formula. With this choice, the proposal distribution and weight update in the algorithm algorithm 1 become:

$$g(x_t|x_{t+1}) \propto p(x_t|x_{t+1})p_\theta(y^*|x_{t+1}), \quad w_t \propto p(x_t|x_{t+1})p_\theta(y^*|x_t)/g(x_t|x_{t+1})p_\theta(y^*|x_{t+1}) \tag{6}$$

Here the proposal $g(x_t|x_{t+1}) = p_\theta(x_t, y^*|x_{t+1})$ is derived by conjugating $\psi(x_{t+1})$ with the transition density, introducing a "guidance term" in the particle update (Wu et al., 2023; Bansal et al., 2024; Moliner et al., 2024).

## 6.3 Revisiting the Blind Forward Problem : Learned Forward Operator $\mathcal{A}$

In the previous section 4, we have approximated $\tilde{\mathcal{A}}[y^*]$ to $\mathcal{A}$ by observing the wet signal $y^*$ generated by $y^* = \mathcal{A}(x^*)$. Then we apply the approximated operator $\tilde{\mathcal{A}}_\theta$ to help to solve the blind inverse problem either. Recall that $\tilde{\mathcal{A}}_\theta$ consists of the reference encoder and the main operator. We train our conditional diffusion model with using the reference encoder as an auxillary condition, and use the main operator in the particle filtering framework.

**Observation Density for a Nonlinear Blind Inverse Problem**   Recent works applying Sequential Monte Carlo (SMC) methods to solve linear inverse problems typically model the observation density $p(y_t|x_t)$ as a Gaussian distribution (Dou & Song, 2024) or partial observation (Cardoso et al., 2023), leveraging the linearity of the operator. In contrast, our approach involves a general non-linear forward operator $\mathcal{A}$, approximated by a neural network $\tilde{\mathcal{A}}_\theta$, a closed-form expression for $p(y_t|x_t)$ is unavailable in general.

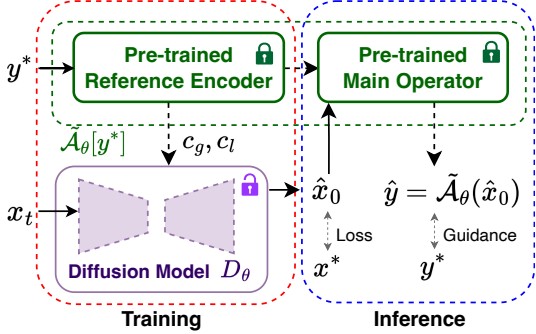

Figure 5: The approximated forward operator $\tilde{\mathcal{A}}_\theta$ is utilized to solve the blind inverse problem. (Red) a conditional diffusion model is trained by $c_g, c_l$ from the reference encoder, and (Blue) $\nabla_{x_t} \tilde{\mathcal{A}}_\theta(\hat{x}_0)$ is calculated every diffusion step.

To address this, we approximate the observation density using the Tweedie's formula as following: (Wu et al., 2023; Chung et al., 2023b; Boys et al., 2023).

$$p(y_t|x_t) = \int p(y_t|y_0)p(y_0|x_0)p(x_0|x_t)dy_0dx_0 \approx \int \mathcal{N}(y_t; \mathcal{A}(x_0), \bar{\sigma}_t)\mathcal{N}(m(x_t), C(x_t))dx_0 \quad (7)$$

where we used $y_0 = \mathcal{A}(x_0)$, and $m(x_t)$ and $C(x_t)$ are the mean and covariance of $p_{0|t}(x_0|x_t)$, presented in Boys et al. (2023). Furthermore, we linearize the operator $\mathcal{A}(\cdot)$ with Taylor expansion around $x_T$, since the integration in Equation 7 has closed-form solution only if $\mathcal{A}$ is linear.

$$\mathcal{A}(x) \approx \mathcal{A}(x_0) + \nabla_x \mathcal{A}(x - x_0) := \mathcal{A}(x_0) + J(x) \quad (8)$$

We further assume that $C(x_t)$ and $J(x_t)$ is small enough compared to $\bar{\sigma}_t$ so that $\bar{\sigma}_t^2 \approx JC(x_t)J^T + \bar{\sigma}_t^2$ to avoid expensive calculation of the gradient of the score function and the operator, which requires $O(T^2)$ complexity where $T$ is a signal length. In conclusion,

$$p(y_t|x_t) \approx \mathcal{N}(y_t; \mathcal{A}(m(x_t)), JC(x_t)J^T + \sigma_t^2) \approx \mathcal{N}(y_t; \mathcal{A}(m(x_t)), \sigma_t^2 I) \quad (9)$$

# 7 EXPERIMENTS

## 7.1 EXPERIMENTAL SETUP

**Dataset**   In our experiments, we train on full-band audio ($\geq$ 44.1kHz) speech datasets, resampled to 44.1kHz. To ensure the forward operator is well-defined, the recording environments of the target and reference signals, denoted as $x, x^*$ in Figure 2, must match in terms of microphone and room characteristics. To achieve this, we organized the training data into two categories, detailed in Appendix D.1. In the *Single Environment* setup, all target and reference pairs were recorded in the same environment; specifically, we used recordings labeled "microphone 1" from the VCTK dataset (Veaux et al., 2017). In contrast, in the *Multiple Environment* setup target and reference pairs are sampled from the same dataset, but different pairs may be drawn from distinct recording environments. Wet audio samples are then generated on-the-fly during training for each dry target and reference audio pair.

**Evaluation Metric**   To evaluate the performance of the proposed method, we use both objective and subjective metrics as follows : 1) **SI-SDR** measures the similarity between predicted and ground-truth wet signals in the waveform domain, 2) **Spectral Convergence (SC) Loss** and **Log-STFT Magnitude (LSM) Loss** in the spectrogram domain, since phase misalignment may not significantly affect perceptual quality, and 3) **Subjective quality** is assessed using Amazon Mechanical Turk (MTurk). Details of the subjective test are in the Appendix F.

## 7.2 BLIND FORWARD PROBLEM : ZERO-SHOT AUDIO EFFECT MODELING

The effectiveness of our method for the blind forward problem is demonstrated via the *zero-shot audio effect learning* task. We generated 100 dry and wet signal pairs for each audio effect type with randomized parameter settings as the test set. Then we report the evaluation metrics together with the subjective test on learning 1) *single-type* audio effects and 2) *complex* audio effects where the forward operator is constructed as described in section 5.

Table 1: Evaluation results for zero-shot audio effect modeling. **Dry** denotes the metric between the dry and wet signals. **Single** and **Multi** refer to the metric between the wet and the predicted signals generated by models trained on single-type and multiple audio effects, respectively. **Diff.P** denotes the subjective score shown by the correct effect but with different parameters.

| | | SI-SDR ↑ | | | SC ↓ | | | LSM ↓ | | | Subjective ↑ | | |
|---|---|---|---|---|---|---|---|---|---|---|---|---|---|
| | | Dry | Single | Multi | Dry | Single | Multi | Dry | Single | Multi | Single | Multi | Diff.P |
| **Noise** | | −3.16 | **12.31** | 10.97 | 1.00 | **0.24** | 0.27 | 1.50 | **0.30** | 0.33 | **61.72** | 61.59 | 26.97 |
| **Filter** | Bandlimiter | −7.45 | **12.26** | 12.17 | 0.80 | **0.20** | 0.20 | 1.54 | **0.24** | 0.24 | **82.71** | 67.22 | 53.10 |
| | Equalizer | 14.84 | 10.80 | 11.61 | 0.40 | 0.22 | **0.20** | 0.59 | 0.25 | **0.23** | 66.97 | **69.93** | 40.52 |
| | Delay | 14.89 | 10.44 | 10.42 | **0.19** | 0.21 | 0.22 | 0.48 | 0.44 | **0.41** | 66.00 | 61.00 | **68.00** |
| | Algo. Reverb | 8.82 | **15.24** | 10.70 | 0.34 | **0.16** | 0.22 | 0.59 | **0.29** | 0.42 | **70.31** | 53.83 | 49.93 |
| | IR Conv | −8.45 | −0.76 | **−0.64** | 0.50 | **0.34** | 0.36 | 1.09 | **0.40** | 0.43 | **66.24** | 65.03 | 62.97 |
| **Nonlinear** | Compressor | 3.01 | **13.50** | 13.19 | 0.60 | **0.21** | 0.23 | 0.96 | **0.28** | 0.31 | 50.28 | **56.28** | 52.28 |
| | Clipping | 4.60 | **23.51** | 21.12 | 0.77 | **0.09** | 0.11 | 2.35 | 0.29 | **0.28** | **67.14** | 62.34 | 58.86 |
| | Distortion | 4.14 | **21.76** | 20.07 | 0.75 | **0.12** | 0.12 | 2.16 | 0.33 | **0.28** | 64.69 | 62.79 | 54.83 |
| **Modulation** | | 2.94 | **13.16** | 9.59 | 0.49 | **0.21** | 0.25 | 0.71 | **0.34** | 0.39 | **50.79** | 48.86 | 53.93 |
| **Codec** | | 9.34 | **19.27** | 17.46 | 0.25 | **0.09** | 0.12 | 1.32 | **0.44** | 0.47 | 56.69 | 57.90 | **65.76** |
| **Multi** | Monolithic | −9.82 | −7.52 | **−0.60** | 0.88 | 0.61 | **0.42** | 1.84 | 0.92 | **0.55** | 50.59 | **71.89** | − |
| | Complex | −10.69 | −4.48 | **0.18** | 0.77 | 0.57 | **0.43** | 1.97 | 0.87 | **0.52** | 52.42 | **68.19** | − |

Table 2: Dependency on the dry signal distribution $\mu$. **Single** and **Multi** refer to the model trained on a single and multiple recording environments, respectively.

| | | SI-SDR ↑ | | | SC ↓ | | | LSM ↓ | | |
|---|---|---|---|---|---|---|---|---|---|---|
| Recording Env. | | Dry | Single | Multi | Dry | Single | Multi | Dry | Single | Multi |
| **VCTK** | Mic 1 | 3.92 | **13.77** | 11.75 | 0.55 | **0.19** | 0.21 | 1.21 | **0.33** | 0.39 |
| | Mic 2 | 4.51 | **13.17** | 12.22 | 0.56 | **0.20** | 0.21 | 1.17 | **0.37** | 0.39 |
| | $\sigma^2 = 0.1$ | -0.35 | **9.97** | 8.84 | 0.64 | **0.23** | 0.26 | 2.61 | 1.83 | **1.81** |
| **DAPS** | | 4.68 | 6.25 | **11.74** | 0.54 | 0.31 | **0.24** | 1.17 | 0.79 | **0.45** |

Two types of models are trained: one exposed only to single-type effects and the other to complex effects during training, and both were evaluated on all effect types.

Results in Table 1 show that our framework successfully replicates general audio effects without prior knowledge of their type. Additional results, including mel-spectrogram comparisons between predicted and wet signals, are provided in Appendix I. Notably, while the single-type model excels in modeling single effects, exposure to complex audio effects during training significantly improves performance on general effects.

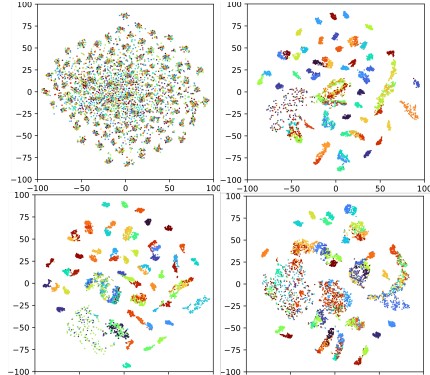

Figure 6: t-SNE of the global conditions $c_g$ from the reference encoder trained on VCTK as $\mu^*$. The top-left shows $c_g$ before training, followed by $c_g$ from VCTK, DAPS, and MAESTRO as $\mu^*$. Each color represents a different forward operator, with $c_g$ extracted from 100 wet audio samples per effect.

### 7.3 SENSITIVITY ANALYSIS ON THE SIGNAL DISTRIBUTIONS $\mu$ AND $\mu^*$

Recall that our model is trained to satisfy $\|\mathcal{A}_\theta[y^*] - \mathcal{A}\|_{L_2(\mu)} \to 0$, implying that the approximation is only guaranteed for $x \sim \mu$. To evaluate sensitivity under distribution mismatch, we trained two models: one trained on VCTK mic 1, and another trained on multiple recording environments. Both models are then tested across different recording environments, including VCTK mic 1 and 2, DAPS, and the VCTK mic 1 perturbed with Gaussian noise with variance 0.1. The DAPS dataset is unseen during training for both models. Results in Table 2 show that both models can approximate the forward operator under distribution mismatch. However, the model trained on multiple environments generalizes better to unseen settings (DAPS) at the cost of performance on seen datasets (VCTK).

We further analyzed cases where the reference signal $\mu^*$ and dry signal $\mu$ are mismatched. Interestingly, a model trained on speech as target signals and piano recordings as references still approximated forward operators (see Appendix B). We attribute this to the global condition $c_g$ encoded by the reference encoder, which appears to capture signal-invariant features. To illustrate, t-SNE visualizations in Figure 6 show that $c_g$ from reference signals $y^*$, generated by the same operator $\mathcal{A}$ but different inputs $x_i$, cluster together.

Table 3: Evaluation results for speech enhancement. **Mix** denotes the metric between the clean and noisy signals. **Appx.** and **GT** denote the metric between the clean and predicted signals generated by models using particles filters of the particle size $N = 4$, with the appoximated operator $\tilde{\mathcal{A}}_\theta$ and the ground-truth operator $\mathcal{A}$. Results for the compressor and the codec are excluded for the GT operator since the compressor returns unstable gradients and the codec is inherently non-differentiable. Audio samples are downsampled to 16 kHz when measure PESQ, eSTOI, and SQUIM

| | | SI-SDR ↑ | | | | PESQ ↑ | | | | eSTOI ↑ | | | | SQUIM-MOS ↑ | | | |
| | | Mix | Cond. | GT | Apprx. | Mix | Cond. | GT | Apprx. | Mix | Cond. | GT | Apprx. | Mix | Cond. | GT | Apprx. |
|---|---|---|---|---|---|---|---|---|---|---|---|---|---|---|---|---|---|
| **Noise** | | 3.64 | 17.52 | 17.37 | **17.86** | 1.40 | 2.54 | **2.55** | **2.55** | 0.66 | 0.77 | **0.78** | **0.78** | 3.08 | 4.25 | 4.13 | **4.27** |
| **Filter** | Bandlimiter | -7.74 | **12.54** | -3.38 | 11.23 | **4.04** | 3.79 | 3.33 | 3.78 | **0.96** | 0.94 | 0.90 | 0.94 | 3.66 | **4.52** | 3.67 | 4.40 |
| | Equalizer | **17.06** | 0.57 | 11.15 | 8.79 | **4.50** | 3.98 | 4.21 | 4.22 | **0.99** | 0.86 | 0.97 | 0.93 | 4.50 | 4.61 | 4.44 | 4.65 |
| | Delay | 15.22 | 9.29 | **18.95** | 15.36 | 2.03 | 2.53 | **2.69** | 2.56 | 0.88 | 0.80 | **0.92** | 0.89 | 3.47 | **4.06** | **4.06** | 4.04 |
| | Algo. Reverb | 8.59 | 15.39 | 15.65 | **16.62** | 1.62 | 2.92 | 2.92 | **2.95** | 0.75 | 0.88 | 0.87 | **0.89** | 3.73 | **4.17** | 4.07 | 4.16 |
| | IR Conv | -5.01 | 7.23 | -1.83 | **8.02** | 3.05 | 3.39 | 3.05 | **3.47** | 0.80 | **0.89** | 0.85 | **0.89** | 3.97 | 4.23 | 3.96 | **4.36** |
| **Nonlinear** | Compressor | 3.75 | **11.76** | - | 11.32 | 3.50 | **3.96** | - | **3.96** | 0.95 | 0.94 | - | **0.96** | 4.58 | 4.59 | - | **4.64** |
| | Clipping | 5.11 | **20.96** | 7.42 | 17.04 | 1.62 | **3.41** | 2.04 | 3.24 | 0.80 | **0.95** | 0.84 | 0.92 | 3.91 | **4.33** | 4.00 | 4.20 |
| | Distortion | 4.63 | **17.81** | 12.23 | 16.80 | 1.61 | **3.28** | 2.49 | 3.22 | 0.80 | **0.94** | 0.90 | **0.94** | 3.50 | **4.32** | 4.26 | 4.26 |
| **Modulation** | | 3.16 | 9.27 | 10.42 | **11.00** | 3.15 | 3.65 | 3.64 | **3.78** | 0.87 | 0.89 | **0.91** | 0.90 | 3.82 | 4.37 | 4.23 | **4.42** |
| **Codec** | | 8.17 | 16.89 | - | **17.20** | 4.10 | 3.52 | - | **3.55** | 0.82 | 0.92 | - | **0.93** | 4.49 | 3.72 | - | **3.82** |

## 7.4 BLIND INVERSE PROBLEM : SPEECH ENHANCEMENT (SE)

We evaluate our method on a *speech enhancement* task, training a conditional diffusion model on the VCTK dataset with a pre-trained reference encoder. During inference, we apply particle filtering with the pre-trained main operator $\tilde{\mathcal{A}}_\theta[y^*]$ and compare the results to those using the ground-truth operator $\mathcal{A}$. An Euler-Maruyama solver with $T = 48$ steps is employed for generation. Notably, our ap-

Table 4: Denoising and Dereverberation results on *VoiceBank/Demand* and *Reverb-WSJ0*.

| Method | SR (Hz) | VoiceBank/Demand | | | Reverb-WSJ0 | | |
| | | PESQ | eSTOI | SI-SDR | PESQ | eSTOI | SI-SDR |
|---|---|---|---|---|---|---|---|
| **Mixture** | – | 1.97 | 0.79 | 8.4 | 1.36 | 0.46 | −7.3 |
| **SGMSE** | 16k | 2.28 | 0.80 | 16.2 | 1.33 | 0.57 | −7.4 |
| **SGMSE+** | 16k | 2.93 | 0.87 | 17.3 | 2.66 | 0.84 | 1.6 |
| **StoRM** | 16k | 2.93 | 0.88 | 18.8 | 2.83 | 0.88 | 6.5 |
| **Our** | 44.1k | 2.45 | 0.82 | 12.3 | 1.46 | 0.51 | -12.3 |

proximated operator is universally applicable without specifying the type of degradation effect, enabling the *universal SE* and including non-differentiable operators such as audio codecs.

Results in Table 3 demonstrate that our approach effectively enhances noisy audio signals across various degradation types. In particular, using the approximated operator during particle filtering even outperforms the ground-truth operator except for delay effect. We hypothesize that when $\hat{x}_0 = D_\theta(x_t)$ is inaccurate due to errors from the diffusion model, the gradient from $\nabla_{x_t}\tilde{\mathcal{A}}_\theta(\hat{x}_0)$ provides the better estimation than $\nabla_{x_t}\mathcal{A}(\hat{x}_0)$. Moreover, twisted particle filtering outperforms the conditional diffusion model according to the Table 3 except for highly non-linear filters like clipping and distortion, due to errors from the linearized operator approximation.

## 7.5 COMPARATIVE STUDIES AND REAL-WORLD SPEECH ENHANCEMENT

Despite training our models only on a single full-band audio dataset (VCTK) with general degradation settings, we evaluate our model on benchmark datasets *VoiceBank/DEMAND* and *Reverb-WSJ0*. We process 1.46 seconds of audio at a time and use an overlap-add method with a 250 ms overlap to handle longer audio signal. we compare our results to baselines : SGMSE (Welker et al., 2022), SGMSE+ (Richter et al., 2023), and StoRM (Lemercier et al., 2023), as shown in Table 4. Although the objective metrics may be lower, the perceptual quality is improved as our model typically extends the audio bandwidth, resulting in perceptually much clean examples. We further provide enhanced samples for real-world noisy speech signals at `https://t.ly/dBUhF`.

## 8 CONCLUSION

We proposed an integrated framework to solve blind forward and inverse problems for zero-shot effect modeling and speech enhancement. For the blind forward problem, we developed a novel framework with a systematic method to generate general forward operators. For the blind inverse problem, we trained a conditional diffusion model and applied twisted particle filtering using the pretrained model from the forward problem. Experiments show that our methods effectively recover both the forward operator and input signal solely from the output signal across various audio effects.

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

## A  CONSTRUCTION OF THE FORWARD OPERATOR

### A.1  CORRESPONDANCE BETWEEN LINEAR DAG AND THE SEMIRING

**Proposition 1** (DAG Representation). *Let $\langle S \rangle$ be a subring of a composition semiring $R$ generated by a subset $S$. Then, each element $\mathcal{A} \in \langle S \rangle$ has a one-to-one correspondence to a linear DAG $\mathcal{G}$, which has a single leaf and root whose nodes are composed of elements in $S$. We refer to such $G$ as a graph representation of $\mathcal{A}$.*

*Proof.* Let $R$ be a composition semiring generated by a set $S$. If $G \in R$, then $G$ can be expressed by the combinations of the finite additions and multiplications of the elements in $S$ by construction. Hence, $G$ is represented by the finite sum of monomials after expansion since the multiplication is distributive. Now enumerate the elements of $S$ according to the order appeared in the monomials so that if $s_j$ comes former than $s_k$ in any monomial $(a_i)$, then $j < k$. Thus we can express as following:

$$G = \sum_{i=1}^{N}(a_i), \quad (a_i) = \prod_{k=1}^{M_i} s_{n_k^{(i)}} \tag{10}$$

where $\{n_k^{(i)}\}$ is a strictly increasing subsequence of the natural numbers, and $M_i$ is the lengths of the monomial $(a_i)$. Then define a chain-shaped graph for the monomial $(a_i)$ by putting nodes as its elements and edges as adjacent multiplication. Formally, $\mathcal{G}_i = (V_i, E_i)$ where a set of nodes $V_i = \{s_{n_k^{(i)}} : n_k^{(i)}, k = 1, ...M_i\}$ and edges $E_i = \{e_{jk} : n_k^{(i)} = n_{j+1}^{(i)}\}$.

Then $\mathcal{G} = (\cup_{i=1}^{N} V_i, \cup_{i=1}^{N} E_i)$ forms a directed acyclic graph with the root $s_m$ and leaf $s_M$ where $m = \min\{n_k^{(i)}\}$ and $M = \max\{n_k^{(i)}\}$.

Conversely, let $\mathcal{G} = (V, E)$ be a directed acyclic graph with one root and leaf. Let $P = \{P_i\} = \{(V_i, E_i)\}_{i=1}^{N}$ be a family of paths from the root and the leaf. Then write a path by multiplication of the nodes by $G_i = \prod_{j=1}^{|P_i|} s_j$ if $s_j \in V_i$. Then $G = \sum_{i_1}^{N} G_i$ is the corresponding element in the semiring we wanted. $\qquad\square$

### A.2  APPROXIMATION THEOREM OF OPERATOR

In this section, we will prove the universal approximation property of the semiring action we constructed at section 5. As stated, the main idea is to imitate the universal approximation theorem of the MLPs. In particular, we corresponds the linear layer of the MLP to the linear operator and activation function to the component non-linear operator.

**Definition 4** (Linear Operator). A function $\mathcal{A} : \mathbb{R}^T \to \mathbb{R}^{T'}$ is a linear operator if it satisfies

$$\mathcal{A}(x + y) = \mathcal{A}(x) + \mathcal{A}(y), \quad \mathcal{A}(ax) = a\mathcal{A}(x), \quad \text{for } x, y \in \mathbb{R}^T, a \in \mathbb{R} \tag{11}$$

Any linear operator $\mathcal{A} : \mathbb{R}^T \to \mathbb{R}^{T'}$ has a matrix representation $A \in \mathbb{R}^{T \times T'}$ such that $\mathcal{A}(x) = Ax$.

Now we consider the following specific type of operators that resemble the structure of the MLP.

**Definition 5** (Ridge Functions). Suppose that $F, L$ are subsets the semiring $R$. Then the set $\mathcal{R}(F, L)$ is a subring of $R$ defined by

$$\mathcal{R}(F, L) = \left\{ \sum_{i \in \mathcal{I}} \sigma^i A^i : x \in X, \sigma^i \in F, A^i \in L \right\} \tag{12}$$

In particular, we choose $L$ to the collection of linear operators $L = \{A^1, A^2, ...\}$ and $F$ to the collection of component non-polynomial non-linear operators $F = \{\sigma^1, \sigma^2, ...\}$. Note that the point-wise operator $\sigma : \mathbb{R}^T \to \mathbb{R}^T$ acts on $x \in \mathbb{R}^T$ by $\sigma((x_1, ...x_T)) = (\sigma_1(x_1), ..., \sigma_T(x_T))$. Concisely we denote this by $[\sigma(x)]_j = \sigma_j(x_j)$ by representing $j$-th coordinate. Then for any input $x \in \mathbb{R}^T$, the action of any element of $\mathcal{R}(F, L)$ can be represented as

$$\left[ \left( \sum_i \sigma^i A^i \right)(x) \right]_j = \left[ \sum_i \sigma^i \left( A^i(x) \right) \right]_j = \sum_i \sigma_j^i \left( \sum_{k=1}^{T} a_{jk}^i x^k \right) \tag{13}$$

where $a^i_{jk}$ is the $j$-th row and $k$-th column of coefficients of the matrix representation of the $i$-th linear operator $A^i$. And $\sigma^i_j : \mathbb{R} \to \mathbb{R}$ is the $j$-th function of the $i$-th component-wise non-linear operator.

The following Lemma is the approximation theorem for the ridge function. We refer to (Lin & Pinkus, 1993; Ismailov, 2020; Pinkus, 1999) for the detail.

**Lemma 1.** *Let $\Omega(\mathcal{U})$ be a subset of all d x n real matrices whose row $(a_1, ...a_n) \in \mathcal{U} = U_1 \times ... \times U_n$ Set*

$$\mathcal{M}(\Omega(\mathcal{U})) = span\{g(Ax) : A \in \Omega, g \in C(\mathbb{R}^n \to \mathbb{R})\} \tag{14}$$

*Then $\mathcal{M}(\Omega)$ is dense in $\mathcal{C}(\mathbb{R}^n, \mathbb{R})$ in the topology of the uniform convergence on compact subsets if and only if a) at least $n - d$ of the $U_1, \ldots, U_n$ have an infinite number of distinct elements; b) at most one of the $U_1, \ldots, U_n$ has only one element, and none has only the zero element.*

*Proof.* See the Theorem 2.1 and Proposition 3.6 of Lin & Pinkus (1993) □

This is the main approximation theorem for our case.

**Theorem 1.** *Suppose that $L = \{A_1, A_2, ...\}$ is a collection of linearly independent linear operators $A_i$ in $C_b(K)$, and $F = \{\sigma_1, \sigma_2, ...\}$ is a set of non-polynomial component-wise continuous functions in $C_b(K)$. Then, for a bounded continuous function, $\mathcal{A} \in C_b(K)$ can be uniformly approximated by the action of $g \in \mathcal{R}(F, L)$.*

*Proof.* Since $F$ are chosen by the collection of component-wise functions, the approximation of $\mathcal{A} : \mathbb{R}^T \to \mathbb{R}^T$ is reduced to $\mathcal{A}_j : \mathbb{R}^T \to \mathbb{R}$ by Equation 13. In the case, $\mathcal{R}(F, L)$ is the $\mathcal{M}(\Omega)$ of the Lemma 1. It suffices to show that our assumptions of $L$ and $F$ satisfies the assumptions of the Lemma 1. However, since $A$ in $C_b(K) : K \to K$, the coefficients of the matrix representation $[a^i]_{jk}$ are bounded in some nonzero compact sets $V_{jk} \subseteq \mathbb{R}$. Choosing $U_k = \cup_{j=1}^T V_{jk}$ to be $U_k$ in the lemma, it satisfies the assumption. Therefore, the approximation is given component-wisely. □

**Remark 1.** Although the operator $\mathcal{A}$ can be approximated using ridge functions, *we do not construct any operator as a ridge function to simulate any forward operator in practice*. This is because the approximation in the theorem assumes a countable sum. Approximations using a finite sum, such as $h(x) = \sum_{i=1}^T \sigma_i(a^i \cdot x)$, are more nuanced, as discussed by Ismailov (2020). Moreover, since our goal is to simulate practical operators for real-world scenarios, sampling pathological operators that heavily distort or erase the content of a speech signal is undesirable and redundant, potentially hindering neural network training.

## B  CROSS DOMAIN RESULTS

Table 5: Effect of the reference: in-domain vs out-of-domain (MASTERO)

| | | SI-SDR ↑ | | | SC ↓ | | | LSM ↓ | | | Subjective ↑ | |
| --- | --- | --- | --- | --- | --- | --- | --- | --- | --- | --- | --- | --- |
| | | **Dry** | **In** | **Out** | **Dry** | **In** | **Out** | **Dry** | **In** | **Out** | **Pred** | **Diff.P** |
| **Noise** | | −3.04 | **9.46** | 6.79 | 1.01 | **0.32** | 0.43 | 1.48 | 0.47 | **0.41** | 70.38 | 39.31 |
| **Filter** | Bandlimiter | −7.43 | **11.21** | 10.05 | 0.79 | 0.21 | **0.20** | 1.54 | **0.36** | 0.47 | 59.71 | 61.75 |
| | Equalizer | **13.46** | 8.68 | 11.36 | 0.43 | 0.25 | **0.21** | 0.59 | 0.47 | **0.27** | 62.69 | 50.21 |
| | Delay | **15.42** | 6.15 | 13.27 | **0.18** | 0.27 | 0.18 | 0.46 | 0.69 | **0.40** | 43.41 | 53.34 |
| | Algo. Reverb | 8.89 | 9.47 | **16.03** | 0.33 | 0.24 | **0.14** | 0.58 | 0.57 | **0.27** | 56.86 | 54.66 |
| | IR Conv | −8.65 | −1.71 | **−1.12** | 0.50 | 0.35 | **0.34** | 1.09 | 0.56 | **0.41** | 58.24 | 63.10 |
| **Nonlinear** | Compressor | 3.24 | 10.24 | **11.27** | 0.61 | 0.27 | **0.25** | 0.95 | 0.48 | **0.41** | 59.28 | 65.10 |
| | Clipping | 4.61 | 21.24 | **24.75** | 0.77 | 0.12 | **0.08** | 2.35 | 0.39 | **0.25** | 71.52 | 60.69 |
| | Distortion | 4.00 | 18.94 | **22.63** | 0.76 | 0.17 | **0.09** | 2.21 | 0.47 | **0.28** | 65.03 | 69.90 |
| **Modulation** | | 3.15 | 10.13 | **15.17** | 0.48 | 0.27 | **0.18** | 0.71 | 0.53 | **0.31** | 57.34 | 59.38 |
| **Codec** | | 9.31 | 17.90 | **19.53** | 0.25 | 0.12 | **0.10** | 1.32 | 0.48 | **0.47** | 52.07 | 53.28 |

# C TRAINING DETAILS

## C.1 ARCHITECTURES

For the reference encoder, we employed the encoder part of the MTFAA-Net. First, it takes the wet reference signal $y^*$ and transform to the STFT domain, and the phase encoder is applied. Then it sequentially downsamples the frequency axis, and each signals are processed by time-frequency conv module and Bi-axial Attention module.

Table 6: Hyperparameters of the architectures. Hyperparameters for the main operators of 1d and 2d models are paranthesized if they are different.

| Reference Encoder | | Main Operator | | Discriminator | |
|---|---|---|---|---|---|
| Parameters | Values | Parameters | Values [1d, 2d] | Parameters | Values |
| Channels | 128 | Channels | [128, 64] | | |
| Channel Mult. | (1, 2, 4) | Channel Mult. | (1, 1, 2, 2, 2) | | |
| Ds Factors | (4, 4, 4) | $c_g$ size | 512 | | |
| Causal | False | $c_l$ size | 512 | | |
| Window Length | 2046 | Self-Attn. | 2 | | |
| Hop Length | 512 | Cross-Attn. | 2 | | |
| | | $n_{res}$ | (2, 2, 4, 4, 4) | | |

## C.2 Audio Effects

All the audio effects are implemented in JAX and operated in CPU with JIT(Just-in-Time) compilation, enabling an on-the-fly generation and rendering of the forward operator. We will also exploit an automatic differentiation system of JAX to calculate $\nabla_x \mathcal{A}(x)$. All the audio effects are implemented based on the algorithms in (Zölzer et al., 2002).

Table 7: AFX Parameter Types

| Class | AFX | Acronym | Parameters (Default Sampling Range) |
|---|---|---|---|
| **2nd Order Filter** | Lowpass | lp | Frequency Hz [1000, 3000], q [0.7, 1.2] |
| | Bandpass | bp | Frequency Hz [250, 5000], q [0.2, 2] |
| | Highpass | hp | Frequency Hz [500, 3000], q [0.5, 4] |
| | Bandreject | brj | Frequency Hz [400, 4000], q [0.2, 2] |
| | Lowshelf | ls | Frequency Hz [200, 3000], q [0.5, 2], Gain dB [6, 18, 9, 6] |
| | Highshelf | hs | Frequency Hz [2000, 7200], q [0.5, 2], Gain dB [6, 18, 9, 6] |
| | Bell | bell | Frequency Hz [120, 3000], q [1, 4], Gain dB [8, 24, 12, 6] |
| | SVF | svf | Frequency Hz [180, 3000], q [0.5, 4], c hp [0.2, 0.8], c bp [0.2, 0.7], c lp [0.2, 0.7] |
| **Ladder** | Bandpass Ladder | bpl | Frequency Hz [700, 4000], k [0, 0.6] |
| | Highpass Ladder | hpl | Frequency Hz [250, 4000], k [0, 0.6] |
| | Lowpass Ladder | lpl | Frequency Hz [800, 3000], k [0.2, 0.6] |
| **Crossover** | Crossover | crs | Frequency Hz [40, 3000] |
| **Memoryless Nonlinearity** | Distortion | dist | Gain dB [8, 32, 12, 6], Hardness [0, 1, 0.5, 0.2], Asymmetry [0, 1] |
| | Hard Clipper | hclp | Gain dB [18, 36, 24, 4] |
| | Soft Clipper | scli | Factor [12, 24] |
| | Bitcrush | bit | Bit Depth [4, 8, 6, 2] |
| **Dynamic Range Controller** | Compressor | cmp | Threshold dB [-24, -6], Ratio [12, 20], Attack. ms [10, 60], Release ms [30, 50], Knee dB [0, 24] |
| | Inverted Comp. | icmp | Threshold dB [-15, -6], Ratio [0.25, 1], Attack. ms [0.1, 50], Release ms [50, 300], Knee dB [0, 24] |
| | Limiter | lim | Threshold dB [-10, -6], Release ms [30, 100] |
| **Modulation Effect** | Chorus | cho | Centre Delay ms [5, 15], Feedback [0.4, 0.7], Mix [0.8, 1, 0.8, 0.1] |
| | Vibrato | vib | Depth [0.5, 1, 0.8, 0.2] |
| | Flanger | fla | Depth [0.5, 1, 0.7, 0.1] |
| | Tremolo | tre | Depth [0.5, 1, 0.7, 0.1] |
| **Delay and Reverb** | Delay | del | Delay Seconds [0.1, 0.3], Feedback Gain dB [-12, -6], Mix [0.4, 0.8, 0.5, 0.25] |
| | Mono Reverb | rvb | Room Size [0.2, 0.8], damping [0.3, 1], Mix [0.3, 0.8, 0.5, 1] |
| | RIR Conv. | rir | |
| | MicIR Conv. | mcir | |
| **Phase Vocoder** | Pitch Shift | pits | Semitones [-12, 12] |
| **Codec** | libopus | lopus | Bitrate [8, 256] |
| | libvorbis | lvobs | Bitrate [48, 200] |
| | aac | aac | Bitrate [8, 256] |

## C.3 Objective Functions

We used SC loss and LSM for the spectogram loss $\mathcal{L}_m$.

- *SI-SDR* (Roux et al., 2018; Luo & Mesgarani, 2017): .

- *Spectral convergence loss (SC), Log-STFT magnitude loss (LSM)* (Yamamoto et al., 2020)

$$\text{SC}(x, \hat{x}) = \frac{1}{N} \sum_{i \in \mathcal{S}} \frac{\||\text{STFT}_i(x)| - |\text{STFT}_i(\hat{x})|\|_F}{\||\text{STFT}_i(x)|\|_F}$$

$$\text{LSM}(x, \hat{x}) = \frac{1}{N} \sum_{i \in \mathcal{S}} \| \log |\text{STFT}_i(x)| - \log |\text{STFT}_i(\hat{x})|\|_1$$

(15)

where $\hat{x}$ represents the predicted signal, and $STFT_i$ denotes the short-time Fourier transform with FFT size $i \in \mathcal{S} = \{2048, 1024, 512, 256\}$ with 75% overlap between windows.

- Subjective: To quantify the perceptual discrepancy between the predicted wet signal and the ground-truth wet signal, we conducted a subjective listening test. The details can be found in the appendix.

## D  EXPERIMENT DETAILS

### D.1  DATASET SPLIT

- **Single Environment**: All target and reference datasets are from the same recording environment. We used the VCTK dataset(Veaux et al., 2017), which has two recording environments. Therefore, we separated this dataset into two sub-datasets and chose one environment for the whole dry and wet audio pair.

- **Multiple Environments**: We use (Bakhturina et al., 2021; Puchtler et al., 2021)

For convolved RIR, we mixed publicly available room impulse datasets for various RIR data.

- Seen Noise : Fonseca et al. (2019) Train set

- Unseen Noise : Fonseca et al. (2019) Valid set

- Seen RIR : Eaton et al. (2016); Jeub et al. (2009); Szöke et al. (2019); Rebecca & Mark (2010); Amengual Gari et al. (2020); Yasuda et al. (2022); Kearney et al. (2022); Traer & McDermott (2016); Dietzen et al. (2023); Murphy & Shelley (2010); Pasoulas et al.; Nakamura et al. (1999); Audio and also used Altiverb, Echotheif, Fokke rir dataset.

- Unseen RIR : Murphy & Shelley (2010) and also used Altiverb, Fokke rir dataset.

- Seen MicIR : Kujawski et al. (2024), and also used Vintage micir dataset.

- Unseen MicIR : Franco Hernández et al. (2022); MICIR

For VCTK, we isolated p231, p271, p311, p347 as a valid set.

## E  PARTICLE FILTERING AND SEQUENTIAL MONTE CARLO

### E.1  SEQUENTIAL MONTE CARLO (SMC) AND TWISTED PARTICLE FILTERING

The goal of Sequential Monte Carlo (SMC) is to estimate $\pi_t(x_{1:t})$ recursively over time. A *target distribution* $\pi_t(x_{1:t})$ is defined by an unnormalized density $\gamma_t(x_{1:t})$ with normalization constant $Z_t$. In the context of state-space models, one of the natural choices is $\pi_t(x_{1:t}) = p(x_{1:t}|y_{1:t})$ by $\gamma_t(x_{1:t}) = p(x_{1:t}, y_{1:t})$ and $Z_t = p(y_{1:t})$. However, as it is typically high dimensional and intractable, direct sampling is unfeasible except for a few cases. Additionally, even if the sampling is feasible, the full trajectory $x_{1:t}$ should be sampled every step $t$ to simulate the target distribution. To address these limitations, *Sequential Importance Sampling (SIS)* is introduced, which enables the sequential approximation of $\pi_t(x_{1:t})$ via importance sampling.

Let $q_t(x_{1:t})$ be a probability density whose support includes that of $\pi_t(x_{1:t})$ and is easier to sample from, referred to as the *importance sampling density*. The importance weight is then defined as the ratio $w_t(x_{1:t}) = \pi_t(x_{1:t})/q_t(x_{1:t})$ and normalized to $\tilde{w}_t$. Given samples $x_{1:t}^i \sim q_t(x_{1:t})$, we can approximate the target distribution and expectations as follows:

Now, assume that $q_t(x_{1:t})$ is factorized as $q_1(x_1) \prod_{k=2}^n q_k(x_k|x_{1:k-1})$. Then, the importance weight can be updated recursively:

$$w_t(x_{1:t}) = \frac{\pi_{t-1}(x_{1:t-1})}{q_{t-1}(x_{1:t-1})} \frac{\gamma_t(x_{1:t})}{\gamma_{t-1}(x_{1:t-1})q_t(x_t|x_{1:t-1})} = w_{t-1}(x_{1:t-1})u_t(x_{1:t}) \tag{16}$$

where $u_t(x_{1:t}) = \gamma_t(x_{1:t})/\gamma_{t-1}(x_{1:t-1})q_t(x_t|x_{1:t-1})$ is the incremental importance weight.This recursive formulation reduces computational complexity by reusing previous weights and particles.

As time progresses, the variance of the weights $w_t^i$ tends to increase, causing *weight degeneracy* where only a few particles carry significant weight. To address this, resampling is performed periodically, replacing low-weight particles with high-weight ones such as systematic, residual, and multinomial resampling.

**Particle Filtering** Now we aim to estimate the marginal filtering distribution $p(x_t|y_{1:t})$ on the state-space model. Note that the distribution has a recursive relation by the prediction step, $p(x_t|y_{1:t-1}) = \int p(x_t|x_{t-1})p(x_{t-1}|y_{1:t-1})dx_{t-1}$ and the update step $p(x_t|y_{1:t}) \propto p(y_t|x_t)p(x_t|y_{1:t-1})$. However, this integration is intractable except for the case of finite SSM or linear Gaussian SSM, where the latter has a tractable solution known as Kalman filter. Therefore, we need an approximation to evaluate the marginal distribution using SMC. By setting $\gamma_t(x_{1:t}) = p(x_{1:t}, y_{1:t})$, so $\pi_t(x_{1:t}) = p(x_{1:t}|y_{1:t})$ and $Z_t = p(y_{1:t})$. Suppose that we have approximated $p(x_{t-1}|y_{1:t-1})$ by $\hat{p}(x_{t-1}|y_{1:t-1}) = \sum_{i=1}^{N} w_{t-1}^i x_{t-1}^i$ by $x_{t-1}^i \sim q(x_t|y_{1:t})$. Then we have

$$\hat{p}(x_t|y_{1:t-1}) = \sum_{i=1}^{N} w_{t-1}^i p(x_t^*|x_{t-1}^i), \quad \hat{p}(x_t|y_{1:t}) = p(y_t|x_t) \sum_{i=1}^{N} w_{t-1}^i p(x_t^i|x_{t-1}^i) \quad (17)$$

with the updating function $w_t^i \propto w_{t-1}^i p(y_t|x_t^i)p(x_t^i|x_{t-1}^i)/g(x_t^i|x_{1:t-1}^i, y_{1:t})$. While the particle filtering reflects the observation sequences up to the current step $t$, we can incorporate the future observation through the twisting function $\psi_t$. By this choice of the twisting function, the prediction and update steps are twisted by $p_t^{\psi}(x_t|x_{t-1}) = p(x_t|x_{t-1})\psi_t(x_t)/\tilde{\psi}_{t-1}(x_{t-1})$ and $p_t^{\psi}(y_t|x_t) = p(y_t|x_t)\tilde{\psi}_t(x_t)/\psi_t(x_t)$. While remaining the terminal target distribution $\pi(x_{1:T}|y_{1:T}) = \tilde{\pi}(x_{1:T}|y_{1:T})$ invariant.

## F    Subjective Test

We use Amazon Mechanical Turk (MTurk) to conduct the subjective evaluation. A total of 30 participants were recruited and assigned to evaluate the audio samples based on the provided instructions. We eliminated 3 participants who did not pass the attention check test, resulting in 27 participants total. To evaluate the perceptual quality of audio transformation, a subjective listening test was conducted using a set of reference and test audio signals. The test follows the below procedure to assess how well transformed (wet) audio resembles the target wet audio, given a reference dry-wet pair.

1. **Reference Listening**: Participants first listen to two reference audio signals:

    - **Dry Reference**: The unprocessed (dry) version of the audio.
    - **Wet Reference**: The processed (wet) version of the same audio, transformed using an audio effects (AFX) mapping.

    These reference signals are provided to inform participants with the transformation effect and the expected result.

2. **Target Listening**: After listening to the reference signals, participants are presented with a new dry target audio signal that has not been processed.

3. **Expectation Formation**: Participants are instructed to imagine the expected wet version of the target audio based on the transformation they heard in the reference signals.

4. **Rating**: Participants are then presented with several test audio samples, each a processed version of the dry target audio, and asked to rate how similar each sample is to the imagined wet target audio using a slider. The rating scale is as follows:

    - **0**: Very poor resemblance to the desired wet target audio.
    - **100**: Identical to the wet reference.

## G    Further Applications

Further applications inhibit in areas such as Automatic Dialog Replacement (ADR), recording environment normalization, automatic mixing and mastering, and timbre transfer. By recovering the dry signal using our inverse problem approach and applying transformations with the forward problem method, we can facilitate tasks like transferring audio characteristics between signals and enhancing overall audio production processes.

## H   LIMITATIONS AND FUTURE WORK

Our study currently focuses on single-input single-output (SISO) systems with fixed signal lengths; extending it to handle multi-input multi-output (MIMO) systems and variable-length signals would enhance versatility. The approach relies on input signals from a known distribution, so performance may degrade with significant deviations—developing robustness to input variations is important. Computational complexity is also a concern, making real-time applications challenging and necessitating efficiency optimizations. Additionally, while effective in audio applications, extending the framework to other domains remains an open challenge, and some theoretical assumptions may not hold universally, requiring further analysis.

## I   ADDITIONAL RESULTS ON FORWARD OPERATOR LEARNING

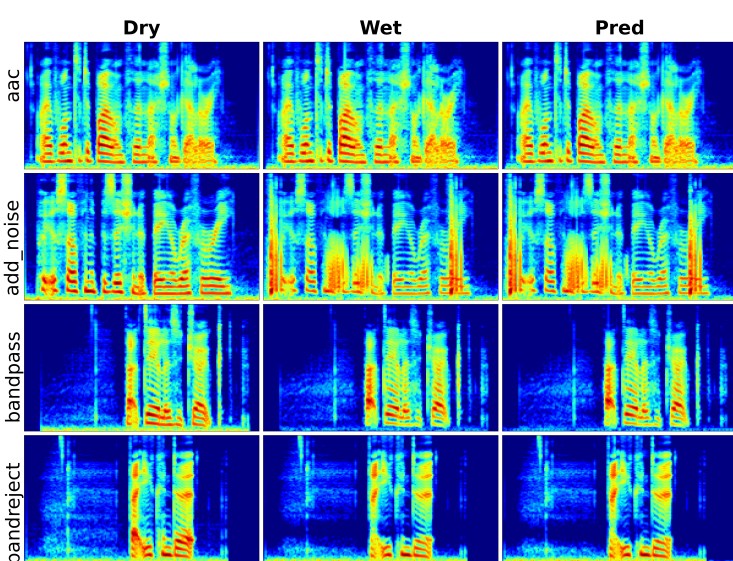

Figure 7: Mel Spectrogram of Single Audio Effect

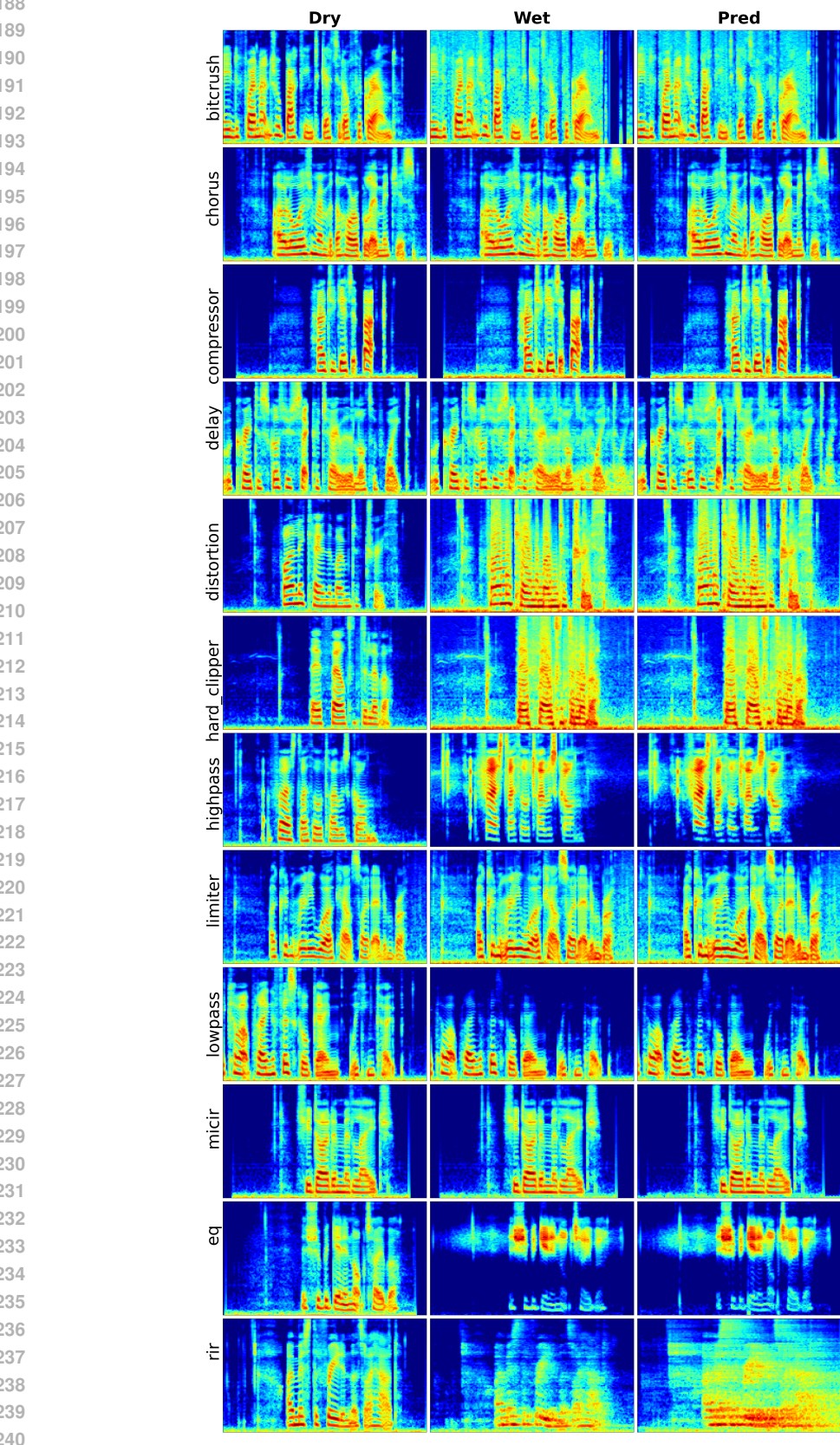

Figure 8: Mel Spectrogram of Single Audio Effect

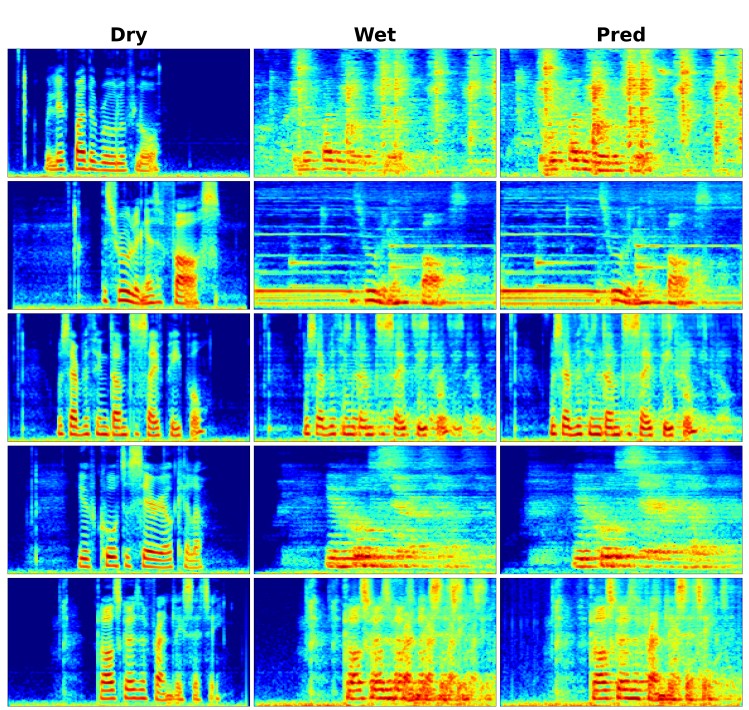

Figure 9: Mel Spectrogram of Monolithic AFX Graph

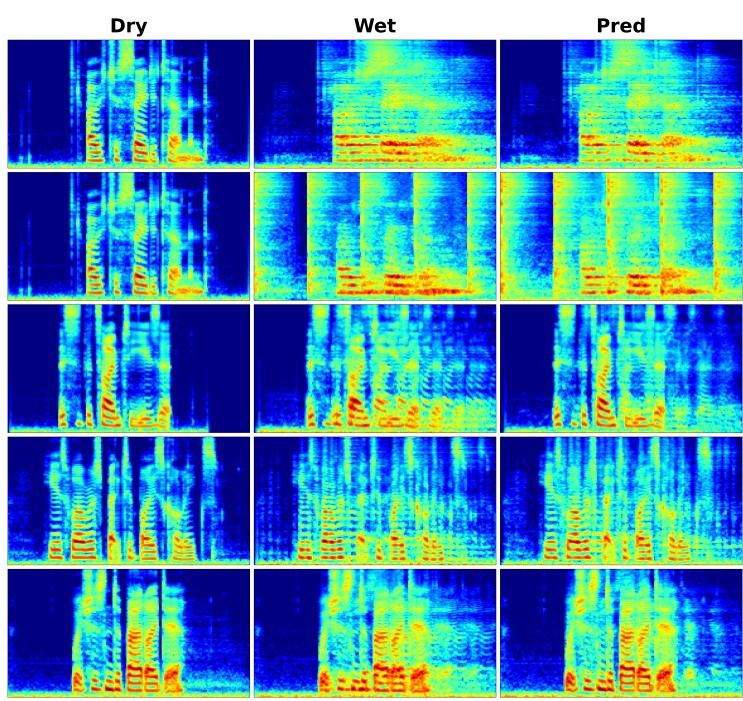

Figure 10: Mel Spectrogram of Complex AFX Graph

