# OpenReview forum: "Solving Blind Non-linear Forward and Inverse Problem for Audio Applications"
_ICLR.cc/2025/Conference — Submitted to ICLR 2025_

### Official Review · Reviewer_xo99 · 2024-10-30

**Soundness:** 1
**Presentation:** 1
**Contribution:** 1
**Rating:** 1
**Confidence:** 4

**Summary:**

This paper proposes a novel framework for solving blind forward and inverse problems using diffusion models. The approach consists of two main steps:

1. Training an encoder network that takes a transformed signal (wet signal) as input and outputs the parameters of the transformation process (forward operator).
2. During inference, using the parameters estimated by this encoder network to solve the inverse problem, with the diffusion model serving as a prior distribution.

The framework is applied to address the following problems:

- Estimating the original signal (dry signal) from a wet signal created by audio effects with unknown parameters.
- Speech enhancement, which involves restoring the original speech signal from degraded speech that has undergone various processing.

This approach enables the solution of inverse problems where the transformation process is unknown, leveraging the power of diffusion models as priors.

**Strengths:**

This research topic is important both theoretically and practically as it explores new applications using diffusion models as a foundation. A particularly noteworthy aspect is the potential to solve blind inverse problems with minimal additional training.

**Weaknesses:**

I find it difficult to recommend this paper for acceptance due to the following reasons. Detailed comments and questions are provided in the Questions part.

1. There are several parts of insufficient explanations. For example, the meaning of 'hallucination effect' on line 57 is unclear, and the caption for Figure 1 is incomplete. Addressing these and other similar issues would improve the paper.
2. While the paper contains a lot of mathematical descriptions, many terms are undefined or seem unnecessary for the discussion. Although some meanings can be inferred, the lack of proper citations makes it challenging for readers to follow the discussion accurately. For instance, A_{\theta}^{\dagger} on line 145 and c_g, c_l on line 161 are not properly defined. Additionally, Proposition 1 and Theorem 1 are stated without proofs.
3. The problem addressed in this paper appears to be inadequately defined, and it's unclear from the paper whether the method described in section 6.2.3 actually solves this problem.
4. A major issue is that the experimental results in Section 7 do not include any comparative methods. This makes it impossible to discuss the performance advantages of the proposed method.

**Questions:**

I would like to request clarification from the authors on the following points:

1. Please provide evidence for the claim near line 49 “that generative models are inferior to discriminative models in terms of quality”.
2. Explain the meaning and relevance of the statement on line 57 that particle filtering techniques reduce the 'hallucination effect'.
3. In Definition 1, if a pure noise-adding operator (as might be expected in speech enhancement tasks) is unbounded, is it excluded from the forward operators considered in this paper?
4. Complete the explanation following '(Right)' in Figure 1.
5. Clarify the definition of 'forward problem' mentioned on line 129.
6. Provide the definition of A_{\theta}^{\dagger} on line 145.
7. Define c_g and c_l mentioned on line 161.
8. Explain what 'sequence length' refers to on line 188.
9. Demonstrate how the training objective in section 4.3 is derived from the objective function defined in section 3.
10. Provide the proofs for Proposition 1 and Theorem 1.
11. Explain how the results of Theorem 1 contribute to the paper's main arguments.
12. Describe the derivation method for m(x_t) and C(x_t) in equation (8) of section 6.4.
13. Can you add comparative methods to section 7?
14. Is it possible to provide audio samples in the experimental section, e.g., audio samples corresponding to the spectrograms shown in Figures 6-9?"

---

> ### Author Response · Authors · 2024-11-25
>
> Thank you for your constructive and kind feedbacks. We appreciate the detailed comments you have made. For the primary revisions, please refer to the general comment we have made.
>
> **Q1.Please provide evidence for the claim near line 49 “that generative models are inferior to discriminative models in terms of quality”.**
> - We have removed this statement, recognizing it was overstated. The original intention was the state-of-the-art speech enhancement on various benchmark test set have been from discriminative rather than generative. However, considering its flexibility and generality, we conclude that the statement is not fair and overstated.
>
> **Q2. Explain the meaning and relevance of the statement on line 57 that particle filtering techniques reduce the 'hallucination effect'**
> - The hallucination effect refers to generative models producing artifacts or unrealistic outputs that deviate from the original signal’s characteristics. Since particle filtering is expected to better approximate the posterior distribution, the enhanced outcome is more coherent to the observed noisy speech, in terms of $y^* \approx \mathcal A(\tilde x)$. However, we removed the statement since we did not quantify or examine the effect on the hallucination effect in detail in the main text.
>
> **Q3. In Definition 1, if a pure noise-adding operator (as might be expected in speech enhancement tasks) is unbounded, is it excluded from the forward operators considered in this paper?**
> - Yes, unbounded operators are excluded. We focus on additive noise with 'finite SNR levels' between -5 and 15 dB, ensuring the operator remains bounded.
>
> **Q4. Complete the explanation following '(Right)' in Figure 1**
> - We have revised all the incompleted statement in the paper. Thank you for pointing it out.
>
> **Q5. Clarify the definition of 'forward problem' mentioned on line 129.**
> - We have elaborated on the definition in Section 3.1, Definition 2. Briefly, the forward problem involves estimating the forward operator $\mathcal{A}$ from the observed signal $y^*$, where $y^* = \mathcal{A}(x^*)$.
>
> **Q6. Provide the definition of A_{\theta}^{\dagger} on line 145.**
> - The notation $\dagger$ was intended to represent the pseudo-inverse of $\mathcal{A}$, such that $\mathcal{A}^\dagger \mathcal{A}(x) = x$. To avoid confusion, we have replaced this notation with $\tilde{\mathcal{A}}_\phi$.
>
> **Q7. Define c_g and c_l mentioned on line 161.**
> - $c_g$ and $c_l$ are global and local conditions from the reference encoder, respectively. We detailed the explanation in the main text.
>
> **Q8. Explain what 'sequence length' refers to on line 188**
> - We represents $c_g$ as a single vector and $c_l$ as a sequence of vectors. The sequence length means the length of the vectors in $c_l$.
>
> **Q9. Demonstrate how the training objective in section 4.3 is derived from the objective function defined in section 3.**
> - The training objective in Section 4.3 minimizes the empirical 2-Wasserstein distance between the push-forward measure of the approximated forward operator and the reference distribution. The additional terms (e.g., adversarial and spectrogram losses) are added to enhance perceptual quality and enforce domain-specific constraints.
>
> **Q10.11. Provide the proofs for Proposition 1 and Theorem 1., result of the theorem 1**
> - We have added sketches and main ideas of the proofs in the main text and Appendix. Theorem 1 validates the flexibility and generalizability of our approach to handle diverse audio effects and degradations, given sufficient 'basic operators' comprising linear and pointwise nonlinear operators.
>
> **Q12. Describe the derivation method for m(x_t) and C(x_t) in equation (8) of section 6.4.**
> - These terms represent the mean and covariance of the posterior distribution $p(x_0|x_t)$. We refer to the results in the paper "Tweedie moment projected diffusions for inverse problems (Boys et al., 2023)" for the explicit derivation. Specifically, the posterior mean $m(x_t)$ is known as Tweedie's formula, equivalent to the denoising function $\hat{x}0 = D\theta(x_t)$ in "Elucidating the Design Space of Diffusion-Based Generative Models" (Karras et al., 2023). $C(x_t)$ is the second moment of the posterior distribution, expressed as the gradient of the score function.
>
> **Q13. Comparative Methods**
> - We have added descriptions of the comparative methods at the end of the main text for the Voicebank/Demand and Reverb-WSJ0 benchmarks. While the objective metrics are worse than the baselines, their perceptual quality is good.
>
> **Q14. Is it possible to provide audio samples in the experimental section, e.g., audio samples corresponding to the spectrograms shown in Figures 6-9?**
> - Yes, we have provided an extensive list of audio samples at the link mentioned in the general comment.
>
> Thank you again for your constructive and detailed feedback.

---

> > ### Comment · Reviewer_xo99 · 2024-11-26
> >
> > Thank you for your thorough response and efforts to address the concerns I raised. While I agree with many of your answers, I still have some remaining concerns. Although I recognize that significant improvements have been made, I would like to maintain my original score.
> >
> > Regarding the novelty of the proposed method:
> > 1. Is the Twisted Particle Filter presented in Algorithm 1 the proposed method? If so, how does it differ from other methods? I understand that the linear approximation of the operator A is unique to this method, but how is this reflected in Algorithm 1?
> >
> > 2. Concerning the evaluation of the proposed method and comparative methods:
> > I recognize that this problem setting itself is challenging, and I understand that it may underperform compared to state-of-the-art methods specifically optimized for this purpose. However, describing the results simply as "their perceptual quality is good" lacks sufficient discussion. (Since metrics like PESQ and eSTOI are supposed to evaluate perceptual quality, such an expression might be inappropriate.)
> >
> > 3. Where is Theorem 1 proven?

---

> ### Author Response · Authors · 2024-11-29
> **Thank you for the valuable responses**
>
> Thank you again for your response to the revised versions. We address the concerns you raised.
>
> 1. Yes, using the **twisted particle filter** with the **approximated operator** ($\tilde{\mathcal A}_\theta$ in our notation) is one of the main methods proposed in this work.
>
> We emphasize the following primary advantages and claim novelties:
> - **Combining and Generalization of the previous works by the approximated operator**
>
> Prior works on diffusion-based inverse problems can be categorized into two groups.
>
> - First, inverse problems with a known operator [1, 2, 3, 4, 5]. These approaches usually factorize $\nabla_{x_t} \log p(x_t|y)$ into $\nabla_{x_t} \log p(x_t) + \nabla_{x_t} \log p(y | x_t)$ and approximate the latter term via "posterior sampling." When dealing with linear operators, some works factorize the matrix $y = Ax$ using Singular Value Decomposition (SVD) to find a subspace for solving the problem. Image inpainting is a primary application in this context since the "masked portion" is known a priori.
>
> - Second, blind inverse problems: There are many works on blind inverse problems; however, they are 'quasi-blind' in terms of specifying the structure of the operator. For example, BlindDPS [6] and BUDDy [7] assume $y = k * x$ and estimate the parameter $k$ during the generation process. More generally, GibbsDDRM [8] assumes $y = H_\phi * x$ to cover general linear operators.
> However, these methods are not applicable in real-world speech enhancement settings, where presuming the kinds or structures of operators is infeasible due to the presence of complex degradation factors such as reverberation, noise, and filters.
>
> Our method both combines and generalizes these prior approaches. By approximating the **general forward operator in a zero-shot fashion** during the generation process, we overcome the limitations of existing methods that rely on known or structurally specified operators. Furthermore, the **'twisting' component of the twisted particle filter corresponds to computing the guidance term** in the first category of methods. Therefore, we assert that our approach is more applicable to universal speech enhancement problems and addresses challenges that have not been tackled by previous works.
>
> - **Application of Particle Filters to Speech Enhancement**
>
> Particle filters operate by running $N$ Markov chains simultaneously and resampling each particle after the prediction step based on their weights, which are closely related to how well each prediction aligns with the observation. We apply this technique to the speech enhancement task to improve the quality of generation—a novel application not explored in similar works.
>
> - **Addressing Non-Linear Inverse Problems**
>  Prior works using twisted particle filters in diffusion process are only dealing with the linear and known cases [8, 9, 10]. However, when dealing with the non-linear operator, the main techniques used in the prior works are not available. Since many degradation operators appeared speech enhancement are highly nonlinear or non-differentiable (audio codec), we address this by simply linearizing operator as treating it as a linear operator by leaving the first term of the Taylor expansion. The linearlized operator is used at sampling from the proposal distribution and calculating weights for each particle in Equation 6. We suggest the future works to incorporate the first derivative term to reduce the approximation error such as low-rank approximation or diagonal approximation suggested in [11].
>
> ---
>
> 2. We agree that merely stating "the perceptual quality is good" is insufficient and not appropriate. The average MOS measured by SQUIM, a reference-less speech quality assessment tool, is 3.77 for the Voicebank/Demand dataset and 4.18 for the WSJ0+Reverb benchmark, even though the objective metrics are poor. Upon reviewing the audio samples we provided, we noticed that the clean audio examples contain inherent noise and are band-limited. Our model attempts to enhance these aspects as well. We will update our results with models trained only on matched datasets for the benchmark to provide a fair comparison.
>
> Again, Thank you for your valuable advice on this matter.
>
> ---
> 3. Sorry for the inconvienience. we mistakenly omitted that part in the appendix. We have updated the proof in the appendix.
>
> We appreciate the reviewers' insightful comments and believe that our revisions address the concerns raised. Thank you for your consideration.

---

> > ### Author Response · Authors · 2024-11-29
> > **Citations for the previous comment**
> >
> > [1] Hyungjin Chung, Jeongsol Kim, Michael Thompson Mccann, Marc Louis Klasky, and Jong Chul Ye. Diffusion posterior sampling for general noisy inverse problems. In The Eleventh Interna- tional Conference on Learning Representations, 2023b. URL https://openreview.net/ forum?id=OnD9zGAGT0k.
> >
> > [2] Bahjat Kawar, Michael Elad, Stefano Ermon, and Jiaming Song. Denoising diffusion restoration models, 2022. URL https://arxiv.org/abs/2201.11793.
> >
> > [3] Jonathan Ho, Tim Salimans, Alexey Gritsenko, William Chan, Mohammad Norouzi, and David J. Fleet. Video diffusion models, 2022.
> >
> > [4] EloiMoliner,MaijaTurunen,FilipElvander,andVesaVa ̈lima ̈ki. Adiffusion-basedgenerative
> > equalizer for music restoration. arXiv preprint arXiv:2403.18636, 2024.
> >
> > [5] Jiaming Song, Arash Vahdat, Morteza Mardani, and Jan Kautz. Pseudoinverse-guided diffusion models for inverse problems. In International Conference on Learning Representations, 2023. URL https://openreview.net/forum?id=9_gsMA8MRKQ.
> >
> > [6] Hyungjin Chung, Jeongsol Kim, Sehui Kim, and Jong Chul Ye. Parallel diffusion models of operator and image for blind inverse problems. In Proceedings of the IEEE/CVF Conference on Computer Vision and Pattern Recognition, pp. 6059–6069, 2023a.
> >
> > [7] Naoki Murata, Koichi Saito, Chieh-Hsin Lai, Yuhta Takida, Toshimitsu Uesaka, Yuki Mitsufuji, and Stefano Ermon. Gibbsddrm: A partially collapsed gibbs sampler for solving blind inverse problems with denoising diffusion restoration. In International conference on machine learning, pp. 25501–25522. PMLR, 2023.
> >
> > [8] Gabriel Cardoso, Yazid Janati El Idrissi, Sylvain Le Corff, and Eric Moulines. Monte carlo guided diffusion for bayesian linear inverse problems, 2023. URL https://arxiv.org/abs/ 2308.07983.
> >
> > [9] Luhuan Wu, Brian L Trippe, Christian A. Naesseth, David M Blei, and John P Cunningham. Practical and asymptotically exact conditional sampling in diffusion models. arXiv preprint arXiv:2306.17775, 2023.
> >
> > [10] Zehao Dou and Yang Song. Diffusion posterior sampling for linear inverse problem solving: A filtering perspective. In The Twelfth International Conference on Learning Representations, 2024. URL https://openreview.net/forum?id=tplXNcHZs1.

---

### Official Review · Reviewer_52hb · 2024-11-01

**Soundness:** 2
**Presentation:** 3
**Contribution:** 2
**Rating:** 5
**Confidence:** 2

**Summary:**

This paper proposes a framework for solving blind forward and inverse problems in audio effect modeling and speech enhancement, where the authors aim to recover the applied audio effect or remove degradations without prior knowledge of the effect chain. The approach includes a dynamic pipeline that generates paired dry and wet signals dynamically and a reference encoder that conditions the network to apply or reverse audio effects. The method is evaluated through objective metrics, with code being released to support reproducibility.

**Strengths:**

- The paper introduces a dynamic signal pairing pipeline that enhances the adaptability of the model across diverse audio effects and environments, a practical contribution to real-time audio processing.
- Using a reference encoder for conditioning allows the model to handle multiple unknown degradations in a versatile manner, making it potentially useful in scenarios requiring "real-time"/zero-shot  adaptability.
- Objective metrics provide an initial evaluation of performance across varied conditions.

**Weaknesses:**

- Theoretical and Practical Limitations: The model employs a Directed Acyclic Graph (DAG) with a semiring-based approach to formalize audio effect chains, which could theoretically add rigor if the properties (like associativity and distributivity) were applied explicitly to audio effects. However, without clear practical benefits, the connection risks remaining overly theoretical. The paper could benefit from further exploration of how this semiring-DAG structure enhances computation or generalization in practice.
- Irreversible Transformations and Restoration Quality: Certain audio effects, such as clipping and band limiting, are irreversible and introduce permanent information loss. While restoration is theoretically possible if the model learns the underlying signal distribution, the quality of restoration is critical and must be rigorously evaluated. In particular, reconstructing fine details - like distinguishing between high-frequency fricative sounds (e.g., "s" and "f") - may be inherently ambiguous after irreversible effects. This limitation should be acknowledged in the framework to set realistic boundaries on reconstruction accuracy.
- Importance of Perceptual Validation: While the objective metrics provide quantitative insight, they are insufficient to fully assess restoration quality, especially in cases involving irreversible transformations. An actual subjective evaluation through listening tests is essential to validate whether the restored audio meets perceptual quality standards, as small deviations from the original can significantly affect listener experience. The lack of subjective tests here limits the confidence in the model’s real-world applicability.
- Limited Comparative Evaluation and Reproducibility: The absence of task-specific baseline comparisons detracts from a clear evaluation of the method’s effectiveness. Additionally, the lack of accessible qualitative examples (e.g., audio samples) limits reproducibility and a deeper evaluation of perceptual quality.
- Result Interpretation: Although the results section provides metrics, a subjective evaluation is missing.

**Questions:**

- How does the proposed method compare to task-specific baselines in terms of objective and perceptual quality?

- Could the model incorporate assumptions about the original signal distribution to enhance the robustness of the restoration, and would this improve restoration quality?

- In the context of semiring theory, could the properties (like associativity and distributivity) of operations in the DAG be explicitly defined to enhance computation or generalization?

- To improve validation, could a listening test be added to evaluate the perceptual acceptability of restored audio, especially in cases where irreversible transformations have introduced ambiguity?

---

> ### Author Response · Authors · 2024-11-25
>
> We sincerely appreciate your advisory and valuable feedback. The most important revisions are presented in the general comments along with the audio samples. Below, we address your main concerns:
>
> **Irreversible Transformations and Restoration Quality**
>
> - As you pointed out, an inverse mapping does not generally exist for the irreversible transformations in a deterministic sense. Therefore, we have defined our inverse problem within a probabilistic framework. Despite the irreversible nature of some transformations, our objective metrics in Tables 1 and 3, along with the perceptual quality of the recovered signals, demonstrate that fine details can be effectively restored.
>
> **Importance of Perceptual Validation, Limited Comparative Evaluation and Reproducibility**
>
> - We have included results from subjective listening tests for the blind forward problem in Table 1. Additionally, we have provided audio samples for each experiment to facilitate perceptual validation. Please refer to the general comments for links and further details.
>
> **Q1. How does the proposed method compare to task-specific baselines in terms of objective and perceptual quality?**
>
> - Unfortunately, task-specific baselines were not examined, as they typically use different training datasets, such as instrumental or music datasets, or operate at lower sampling rates (e.g., 16 kHz). As a result, we found direct comparisons to be unfair. Instead, we evaluate various configurations of our proposed method and provide audio samples to facilitate qualitative assessment of the results.
>
> **Q2. Could the model incorporate assumptions about the original signal distribution to enhance the robustness of the restoration, and would this improve restoration quality?**
>
> - Our framework already leverages a probabilistic formulation, where the original signal distribution corresponds to the distribution of the input signals. This probabilistic approach inherently incorporates assumptions about the signal distribution, contributing to the model's robustness in restoration tasks. While we trained our model on a single dataset and tested it on synthetic data, we acknowledge that training on multiple and more diverse datasets can further improve robustness and generalization, especially in real-world speech enhancement scenarios.
>
> **Q3. In the context of semiring theory, could the properties (like associativity and distributivity) of operations in the DAG be explicitly defined to enhance computation or generalization?**
>
> - We have updated Section 3 to emphasize the advantages of constructing arbitrary forward operators. Properties such as associativity and distributivity allow the forward operator to be rendered in polynomial time by factorizing terms. Using a Directed Acyclic Graph (DAG) representation, the input signal can be processed efficiently via a topological sort. To ensure clarity and avoid excessive theoretical complexity, we have restated the relevant terminologies with concise and sufficient explanations.
>
> **Q4. To improve validation, could a listening test be added to evaluate the perceptual acceptability of restored audio, especially in cases where irreversible transformations have introduced ambiguity?**
>
> - We provide audio samples. Please refer to the general comment to find the url.
>
> Thank you again for your constructive comments.

---

### Official Review · Reviewer_SWe4 · 2024-11-03

**Soundness:** 3
**Presentation:** 3
**Contribution:** 2
**Rating:** 6
**Confidence:** 3

**Summary:**

This paper presents a framework for solving blind nonlinear forward and inverse problems with applications to audio. Specifically, the authors propose using a reference encoder module to approximate an arbitrary forward function for the forward nonlinear problem. Then, based on the estimated forward operator, they propose solving the inverse problem using guidance from the estimated forward function to train the diffusion model for recovering the input signal. Improvements to the main framework are also discussed by using a particle filter-based approach for the inverse problem. Experimental results on audio effects modeling for the forward problem and speech enhancement for the inverse problem are presented to demonstrate the effectiveness of the proposed method, with both objective and subjective evaluations.

**Strengths:**

- The proposed framework is theoretically grounded and can be applied to solving both nonlinear forward and inverse problems.

- Robust performance to distribution shift and recording environment mismatch of the training and test audio data; enabling zero-shot audio effect learning.

- Effectiveness of the proposed method across a wide range of audio effects and manipulations, supported by both objective metrics and subjective listening testing scores.

**Weaknesses:**

- The main weakness of the work is the lack of comparison with existing approaches for better understanding of where the proposed method stands. For example, for the inverse problem of speech enhancement, the authors could have compared their approach to other diffusion based speech enhancement methods (e.g., *Lu et al., "Conditional Diffusion Probabilistic Model for Speech Enhancement," ICASSP, 2022*; *Tai et al., "DOSE: Diffusion Dropout with Adaptive Prior for Speech Enhancement," NeurIPS, 2023*) on more standard benchmark datasets such as VoiceBand+DEMAND (*Cassia et al., "Noisy speech database for training speech enhancement algorithms and TTS models," 2016*) and CHiME-3 (*Vincent et al., "An analysis of environment, microphone and data simulation mismatches in robust speech recognition," Computer Speech Language, 2017*). Having such comparison to prior works will help strengthen the contribution of this paper.

- Another weakness is the lack of information on the model size, computational complexity, or real-time processing performance of the proposed model. From a practical perspective, many speech and audio processing applications have certain constraints on the latency and memory requirements for the models to be deployed on edge devices (e.g., smart speakers, intelligent home appliances, hearing aids, etc). Therefore, providing relevant information would be helpful for researchers and developers to consider adopting the proposed model.

**Questions:**

- In Definition 1 (Forward Operator), it is assumed that both $x\in K$ and $y\in K$, where $K\subseteq X=\mathbb{R}^T$. Does it mean that the proposed framework is only applicable to the case where the input $x$ and output $y$ have the same signal length $T$? Could you provide further details on how the proposed framework can be potentially extended to other inverse problems where the input and output signals may have different dimensionality?

- In eq. (1), what does $\mathcal{A}\in\mathcal{C}(K)$ mean?

- At line 159, maybe directly using $c_g$ for $t$ and $c_l$ for $c$ makes it more clear as they refer to the same thing. This could also help the reader understand what $c_g$ and $c_l$ are in the subsequent sentence at line 161.

- In Figure 2, what are the upper-case $X$ and $Y$? It looks like they are the spectrogram representations of $x$ and $y$, respectively, but such information is missing. In addition, do you use the complex-valued spectrograms or just their magnitude components for training the model?

- In Table 3, what do "Mix", "Blind" and "Known" stand for respectively?

---

> ### Author Response · Authors · 2024-11-25
>
> We sincerely appreciate your advisory and valuable feedback. The most important revisions are presented in the general comments along with the audio samples. Below, we address your main concerns:
> - We have added comparisons with prior works in the experimental section and discussed them accordingly. However, since our model is trained exclusively on full-band audio from the VCTK dataset, it exhibits different enhancement behaviors. Nevertheless, the perceptual quality of our results remains comparable.
> -  As you suggested, we have added detailed information about the model in the appendix. Furthermore, we have publicly released the code to encourage further studies using our proposed method.
>
> **Q1. In Definition 1 (Forward Operator), it is assumed that both $\mathbf{x} \in \mathbb{R}^N$ and $\mathbf{y} \in \mathbb{R}^N$, where $N$ is the signal length. Does it mean that the proposed framework is only applicable to the case where the input $\mathbf{x}$ and output $\mathbf{y}$ have the same signal length $N$? Could you provide further details on how the proposed framework can be potentially extended to other inverse problems where the input and output signals may have different dimensionality?**
>
> A1. Yes, in our work, we assume that both the input signal $\mathbf{x} \in \mathbb{R}^N$ and the output signal $\mathbf{y} \in \mathbb{R}^N$ have the same fixed dimensionality $N$. This assumption simplifies two key aspects:
> - With fixed input and output dimensions, the composition of operators is well-defined without the need to consider varying signal domains. This avoids cumbersome theoretical considerations related to changes in dimensionality during operator composition.
> - We utilize U-Net as our main architecture, which is designed for inputs and outputs of the same size without padding. This choice simplifies the implementation and training process.
>
> To extend our framework to the problems where the input and output signals have different dimensionalities, we need to address these challenges. We suggest some ideas to address these obstacles as following:
> - Instead of using U-net that require fixed input and output sizes, we could employ sequential modeling techniques capable of handling variable-length sequences such as Transformer, VQ-VAE, RQ-Transformer variants, or state-space model (e.g. S4). based on the neural codec (e.g. Encodec)
> - We may track the dimensionality every time the functions are composed. If we only consider the single-type effect, we can just address the problem by the architectural improvement.
>
> **Q2. In eq. (1), what does  $\mathcal A \in C(K)$ mean?**
>
> A2. It denotes $\mathcal A$ belongs to the continuous function from $K$ to $K$. We apologize for any confusion caused by this notation and have updated the main text to clarify this point.
>
> **Q3. At line 159, maybe directly using $\mathbf{x}$ for clean signal and $\mathbf{y}$ for processed signal makes it more clear as they refer to the same thing. This could also help the reader understand what $\mathbf{x}$ and $\mathbf{y}$ are in the subsequent sentence at line 161**
>
> A3. We updated that point following your suggestion.
>
> **Q4. In Figure 2, what are the upper-case $X$ and $Y$?It looks like they are the spectrogram representations of $x$ and $y$, respectively, but such information is missing. In addition, do you use the complex-valued spectrograms or just their magnitude components for training the model?**
>
> A4. In Figure 2, the uppercase letters $\mathbf{X}$ and $\mathbf{Y}$ represent the spectrograms of $\mathbf{x}$ and $\mathbf{y}$, respectively. We apologize for not stating this explicitly and have updated the caption accordingly.
> We use a dual-domain architecture that processes both the waveform and its spectrogram. A 1D U-Net handles the waveform, and a 2D U-Net processes the complex-valued spectrograms with two channels (real and imaginary parts). The outputs from both are combined before producing the final result. This approach captures effects more effectively since some are easier to detect in one domain than the other. For example, "Clipping" appears as simple thresholding in the time domain but introduces non-harmonic structures in the frequency domain. Our primary experiments showed that combining information from both domains improves performance.
>
> **Q5. In Table 3, what do "Mix", "Blind" and "Known" stand for respectively?**
>
> A5. We have updated Table 3 and its caption to clarify the terms. Briefly :
> - (Mix) : The baseline metric between the clean and noisy (mixture) signals.
> - (Cond.): A conditional diffusion model without particle filtering.
> - (GT): A conditional diffusion model applying particle filtering with the ground truth (GT) forward operator during inference.
> (Approx.): A conditional diffusion model applying particle filtering with the approximated forward operator.
>
> Thank you again for your constructive feedbacks !

---

> ### Comment · Reviewer_SWe4 · 2024-11-27
> **Official Comment by Reviewer SWe4**
>
> Thanks to the authors for responding to my questions. My concerns have been partially addressed. I appreciate the authors' effort on revising the manuscript which has clearly improved from the initial version. The audio samples provided and added comparison to other speech enhancement baselines are also helpful. Based on that, I increased my score from 5 to 6. Still, more detailed complexity analysis is missing, which I suggest the authors to include in their future iterations of the paper.

---

> > ### Author Response · Authors · 2024-11-29
> >
> > Dear Reviewer,
> >
> > We are glad that our revisions were helpful in better illustrating our contributions. It's thanks to your valuable feedback that we were able to strengthen our core ideas. We also appreciate your recognition of our efforts and for raising your score from 5 to 6.
> >
> > We acknowledge that a more detailed complexity analysis would enhance our work. In future iterations of the paper, we will incorporate a comprehensive analysis, especially focusing on the evaluation results, as you suggested.
> >
> > Thank you again for your time and consideration.

---

### Official Review · Reviewer_1VZY · 2024-11-04

**Soundness:** 2
**Presentation:** 1
**Contribution:** 2
**Rating:** 1
**Confidence:** 5

**Summary:**

The paper proposes a framework for blind estimation of a forward operator and solving the corresponding inverse problem with application in audio signal transformation. The forward operator is estimated using a neural model consisting of a reference encoder and a “main” neural operator. The inverse problem is solved using a score-matching diffusion model. The authors show results on audio effect learning and speech enhancement.

**Strengths:**

Training pipeline including creating a DAG for the forward operator is interesting, and appears to model more complex forward operators than used in the previous work, e.g., (Rice, 2023).

**Weaknesses:**

The paper proposes a model for estimating the forward operator, and a score-based model for solving the corresponding inverse problem.
The most relevant existing work is (Rice, 2023). However, that is mentioned only in passing, and the difference or the advantages of the proposed model are not clearly presented. Furthermore, there's not a single baseline system used in the experimental section.
This reviewer would suggest, at minimum, to include (Rice, 2023) as a baseline. Furthermore, it would be helpful to include some of the recent diffusion-based speech enhancement models as baselines in the speech enhancement experiment (for example, score-matching based SGMSE).

Surprisingly, for a paper dealing with estimating audio, the authors found not provide a single audio example.
At minimum, the authors should provide randomly-sampled test example pairs of input and output audio (for different processing setups).
A link to the examples should be provided in the paper.

The paper is unfinished and clearly not well prepared for peer review.
For examples, Tables {2,3,4} are not referred to in the paper.
Footnote 2 on page 10 is particularly interesting, where the authors claim that:
"The numbers of the table are relying on the model with insufficient training iteration, thus will be updated during reviewing process."
In this reviewer's opinion, submitting incomplete results is unacceptable.
The authors should have completed the experiments before the initial submission, and all tables should have been properly referenced.

**Questions:**

"Dry and wet signals" naming is used on page 1, but never formally introduced. While that's a common naming used in audio effects, it's not the ideal choice when handling, for example, speech enhancement. The authors could change that to make it more clear. Using general terminology, such as model input and model output or estimated signal, would be more appropriate.

---

> ### Author Response · Authors · 2024-11-25
>
> We sincerely thank you for the advisory and valuable feedback. Most important and primary revisions are presented in the general comment with the audio samples. Below, we respond to your main concerns:
>
> - While we acknowledge similarities between our approach and Rice (2023) in detecting and sequentially removing audio effects from output signals, there are key differences:
>     - Detecting audio effects via classification can lead to degeneracy issues, where different effect combinations result in the same effect, and the classification model may treat alternative combinations as errors. Our approach circumvents this problem by modeling the entire forward operator as a functional form. We have elaborated on this in Section 5 as per your suggestion.
>     - Rice (2023) addresses a finite set of effect compositions with a static, monolithic configuration, whereas our method accommodates a significantly broader range of effect combinations.
>     - RemFX employs effect-specific removal models, limiting flexibility in handling diverse real-world scenarios—a limitation our approach overcomes.
> - We attempted to compare our model's performance with RemFX as recommended. However, direct comparisons are infeasible due to differences in training datasets. RemFX focuses on audio effect removal for instrument datasets (e.g., VocalSet, Guitar Set, DSD100, IDMT-SMT-Drums), while our work centers on speech datasets. Instead, we provide comparisons on standard speech enhancement benchmarks. The audio samples on the benchmark test sets are in the link https://t.ly/dBUhF.
> - As suggested, we have removed the terms "dry" and "wet" signals in Section 1 to avoid introducing terminology prior to their formal definition in Section 2.
>
>  Thank you again for your constructive feedback.

---

> > ### Comment · Reviewer_1VZY · 2024-12-03
> >
> > Thank you for updating the paper.
> >
> > > Detecting audio effects via classification can lead to degeneracy issues, where different effect combinations result in the same effect, and the classification model may treat alternative combinations as errors. Our approach circumvents this problem by modeling the entire forward operator as a functional form.
> >
> > This is a fair point.
> >
> > > Rice (2023) addresses a finite set of effect compositions with a static, monolithic configuration, whereas our method accommodates a significantly broader range of effect combinations.
> > > RemFX employs effect-specific removal models, limiting flexibility in handling diverse real-world scenarios—a limitation our approach overcomes
> >
> > This is a notable difference. However, your approach is evaluated on single-effect configurations. There's no reason not to compare to other works, such as (Rice, 2023), especially since their code is available online.
> >
> > > Instead, we provide comparisons on standard speech enhancement benchmarks.
> > > The audio samples on the benchmark test sets are in the link https://t.ly/dBUhF.
> >
> > Evaluation provided in Section 7.5/Table 4 is not very convincing.
> > Performance of the proposed model is worse than baselines in all conditions, and this is also audible from the audio examples.
> > The authors claim that "Although the objective metrics may be lower, the perceptual quality is improved".
> > This is not the case, especially not so for the reverberant examples. In the reverberant condition, the proposed model performs poorly, which can be verified by listening (note: I accessed these audio examples before the link expired).
> > The statement that the model is better does not seem to be supported by evidence.
> >
> > Also, note that the shortened link expired.
> >
> > > (In updated Section 7.4):
> > > Notably, our approximated operator is universally applicable without specifying the type of degradation effect, enabling the universal SE and including non-differentiable operators such as audio codecs.
> >
> > Based on the results in Table 3, there is some inconsistency in codec results.
> > In particular, both PESQ and SQUIM-MOS are quite high for the input signals ("Mix"), and the proposed model is significantly degrading the quality. At the same time, the proposed model is significantly improving SI-SDR and eSTOI. The differences are so large, that they do not look convincing without audio examples (and the link expired).
> >
> >
> > Overall, the paper is still not suitable for acceptance.
> > The authors did improve the paper but there are several open questions, as could be expected from a submission that was in an rather poor initial state.

---

### Author Response · Authors · 2024-11-25
**General Comment for the Revision**

We sincerely thank the reviewers for their insightful and constructive feedback, which has greatly helped us improve the quality, clarity, and rigor of our paper. Below, we summarize the primary revisions made in response to the reviews:

- Improved Readability and Highlighted Revisions: We revised the main text while maintaining the original core ideas to enhance readability and coherence. Key changes are highlighted in blue for ease of reference.
- Audio Samples for Validation: To facilitate perceptual validation, we present the exhaustive audio samples from our experiments, which evaluated effect-wise, publicly available at https://t.ly/dBUhF.
- Terminology and Problem Formulation (Section 3): We have refined the definitions of key terminologies and reformulated our problem in a probabilistic framework. The definition of the forward operator is now more concise, accompanied by illustrative examples. We aligned the descriptions of the forward and inverse problems, emphasizing the benefits of a probabilistic perspective.
- General Construction of Forward Operators (Section 5): We introduced a general method for constructing arbitrary forward operators in an algebraic manner. While keeping the discussion practically oriented, we highlighted its advantages in real-world scenarios and briefly connected this construction to directed acyclic graphs (DAGs) and approximation results.
- Blind Inverse Problem via Diffusion Models (Section 6): We presented a novel approach to solving the blind inverse problem using a diffusion-based model, enhanced by particle filtering. Viewing the diffusion process as a state-space model conditioned on observations, we developed an algorithm within this framework.
- Reuse of Pretrained Models (Figure 5): We clarified how pretrained models from the forward problem are reused in the inverse problem. Specifically, we employed the reference encoder to train a conditional diffusion model and utilized the approximated forward operator during generation. Unlike prior works, we leveraged the relationship between clean and noisy data to reconstruct clean signals.
- Expanded Experimental Results (Section 7): We elaborated on the discussion of experimental results and included comparative studies to contextualize our contributions. Since our model operates at 44.1kHz and is trained on the VCTK dataset, we invite reviewers to evaluate the perceptual quality of the enhancements on benchmark datasets through the provided audio samples.

We hope these revisions address the reviewers' concerns and significantly improve the paper's overall quality and impact. Thank you again for your thoughtful and detailed feedback.

---

### Meta-Review · Area_Chair_TgYb · 2024-12-20

**Metareview:**

This paper considered the blind inverse problem in audio applications and proposed a unified framework for both the blind forward and inverse problems.

Overall the idea is interesting but most reviewers, including me, agree that this paper is below the acceptance level, due to the following main concerns:

1. The comparisons in the experimental part is insufficient. This is the biggest and widely recognized concern. In the original submission, no comparison was provided. While in the rebuttal the authors provided some, as pointed out by the reviewers, they are still far from sufficient and convincing.

2. This paper is poorly written. The problem formulation and also the method introduction is not well illustrated.

3.  As also pointed out by Reviewer 1VZY, the experiments parts, including the dataset used, are not convincing and there is also some inconsistency.

As such, I cannot recommend acceptance in its current state. I sincerely encourage the authors to revise this paper based on the feedback.

**Additional Comments On Reviewer Discussion:**

The authors addressed some of the concerns but still many concerns remain.

Reviewer 1VZY kept the original score 1, arguing that his/her main concern of insufficient comparison has not been addressed. Also, evaluation provided in Section 7.5/Table 4 is not very convincing, and in Table 3, there is some inconsistency in codec results.

Reviewer SWe4 raised score from 5 to 6, but still concerned the lack of detailed complexity analysis.

Reviewer xo99 recognized the improvements in the rebuttal but still have sincere concerns on the novelty so he/she remains the score.

---

### Decision · Program_Chairs · 2025-01-22

Reject